# Transnational conservation to anticipate future plant shifts in Europe

Yohann Chauvier-Mendes [1,2] ✉, Laura J. Pollock [3], Peter H. Verburg[1,4], Dirk N. Karger [1], Loïc Pellissier [1,2], Sébastien Lavergne [5], Niklaus E. Zimmermann [1,2,6] & Wilfried Thuiller[5,6]

To meet the COP15 biodiversity framework in the European Union (EU), one target is to protect 30% of its land by 2030 through a resilient transnational conservation network. The European Alps are a key hub of this network hosting some of the most extensive natural areas and biodiversity hotspots in Europe. Here we assess the robustness of the current European reserve network to safeguard the European Alps' flora by 2080 using semi-mechanistic simulations. We first highlight that the current network needs strong readjustments as it does not capture biodiversity patterns as well as our conservation simulations. Overall, we predict a strong shift in conservation need through time along latitudes, and from lower to higher elevations as plants migrate upslope and shrink their distribution. While increasing species, trait and evolutionary diversity, migration could also threaten 70% of the resident flora. In the face of global changes, the future European reserve network will need to ensure strong elevation and latitudinal connections to complementarily protect multifaceted biodiversity beyond national borders.

In line with the United Nations Biodiversity Conference of the Parties (COP15) and the recent adoption of the Kunming-Montreal Global Biodiversity Framework (30×30 target), the European Union (EU) seeks to implement a coherent and resilient transnational nature protection network by 2030 covering at least 30% of the land of the EU. The 'EU Biodiversity Strategy for 2030'[1,2] specifies the necessity to improve the European reserve network by further implementing transboundary protected areas to effectively preserve biological biodiversity and nature's contributions to people (NCPs) under global change. Central to this conservation network, the European Alps are one of the largest semi-natural areas of the continent and a centre of plant diversity and endemism[3,4]. Spread across seven countries, the Alps host ~4,500 vascular plant species—more than a third of the flora recorded in Western Europe—with around 400 endemic species[3], and unveil a long history of land use and

geographical processes that has shaped evolutionary and phenotypic plant adaptations over time[5,6].

Alpine and mountain ecosystems are altered by global change in a complex way[7–12]. Many species are expected to migrate upwards increasing the risk of extinction for cold-adapted alpine plants, which have limited colonization opportunity and are potentially suffering from competitive exclusion[13–15]. In the European Alps, not only climate, but also land-use change, is expected to affect this species redistribution, as agricultural land abandonment at high elevation and human activities such as intensification in the lowlands negatively impact mountain biodiversity in Europe[5,16]. While protected areas (PAs) are static entities that aim to preserve biological biodiversity, their networks in mountain ecosystems are generally known to be biased towards higher elevations[17,18], with endangered species sometimes anticipated to naturally migrate within these networks[19]. Therefore, the

[1]Swiss Federal Research Institute (WSL), Birmensdorf, Switzerland. [2]Department of Environmental Systems Science, Eidgenössische Technische Hochschule (ETH) Zürich, Zürich, Switzerland. [3]Department of Biology, McGill University, Montreal, Canada, Quebec. [4]Environmental Geography Group, Institute for Environmental Studies, Vrije Universiteit, Amsterdam, Netherlands. [5]Laboratoire d'Ecologie Alpine, LECA, CNRS, Univ. Grenoble Alpes, Univ. Savoie Mont Blanc, Grenoble, France. [6]These authors jointly supervised this work: Niklaus E. Zimmermann, Wilfried Thuiller. ✉e-mail: yohann.chauvier@wsl.ch

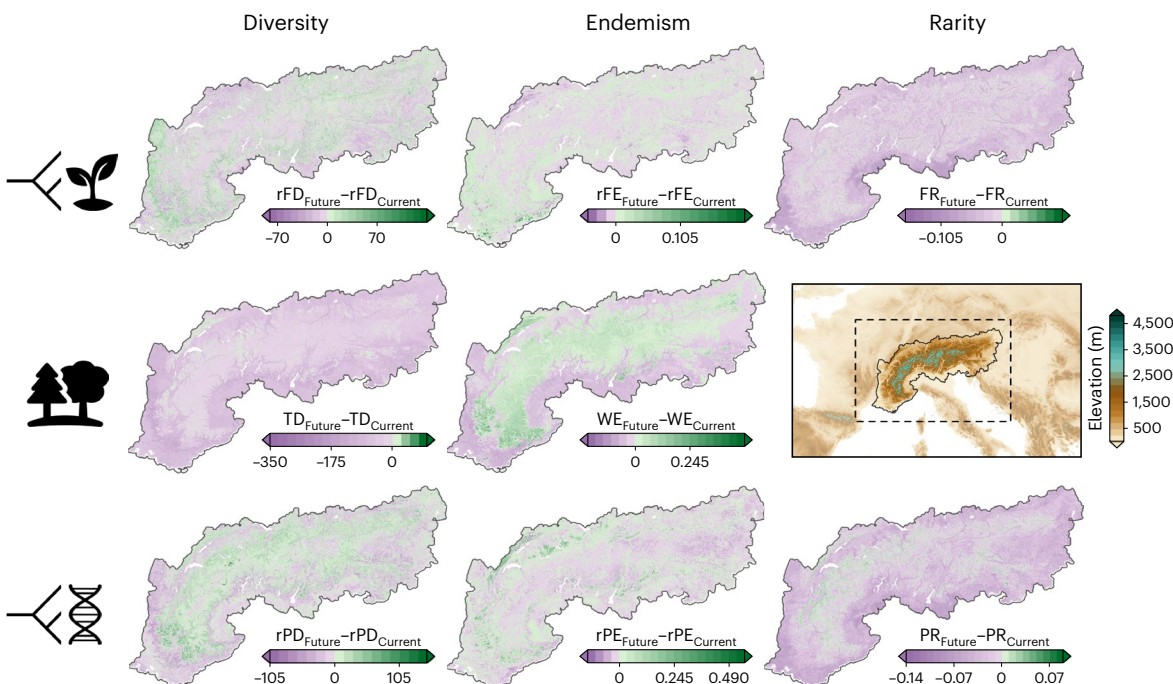

**Fig. 1 | Change in multifaceted diversity and uniqueness by 2050 for SSP245, considering limited plant dispersal.** The first, second and third rows depict the functional, taxonomic and phylogenetic dimensions, respectively. Spatial gains are shown in green, and spatial losses are shown in purple. Relative diversity represents the diversity expected under a given taxonomic diversity (Methods). Silhouettes from the Noun Project.

effectiveness of PAs is also dependent on climate change and changes in land use affecting natural areas, inside and outside these PAs. In this context, we must quantify how species are likely to migrate under global change[20], which species will become threatened and how the current European conservation network should be transnationally adapted to future species range shifts and local extinctions in the European Alps.

Climate and land-use change are expected to influence the facets of biodiversity in different ways, requiring a multidimensional approach to conservation[19,21,22]. Biodiversity is not only about individual species, but is also about 'diversity'—how many species are found in an area or conservation unit (species richness), how much evolutionary history and resilience to environmental changes are shared among these species (phylogenetic diversity)[23] and how diverse their morphological traits and roles in nature are (functional diversity)[24]. From a conservation planning perspective, it is useful to also consider how each local area contributes to the unique biodiversity of the region (for example, species or functions not found elsewhere), which can be measured as species[25], phylogenetic[26] or functional endemism[27]. Finally, 'rarity', which estimates scarcity of unique traits[28] or phylogenetic branches[29], also contributes to 'diversity' and deserves to be considered. It is therefore crucial that conservation planning embraces all these facets to optimize protection complementarity and irreplaceability between geographic areas.

Here we investigate the efficiency of the current European network, and of its potential transnational expansion, in protecting the plant multifaceted diversity and uniqueness of the European Alps at present and for the 2050 and 2080 horizons, under biologically informed (limited) dispersal, two shared socioeconomic pathways (SSPs), seven global circulation models and two land-cover (LC) change scenarios (Methods and summary workflow in Extended Data Fig. 1). Using an ensemble of species distribution models (SDMs) for 1,711 plant species at 100 m resolution, a high-coverage database of species traits and two mega-phylogenies, we predicted future changes in species distributional range, multifaceted diversity and uniqueness in the study area. Using systematic conservation planning (SCP), we

then identified conservation priorities for all types of diversity, areas where these priorities are stable into the future (overlap in current and future priorities) and areas that are critical for expanding the current PAs to meet the 2030 targets while achieving a resilient and effective reserve network.

## Results and discussion
### Upward shifts of multifaceted diversity
Overall, our SDM approach showed very good performances, with an average true skill statistics and Boyce index of the kept models ranging between ~0.6 and ~0.8 across the 1,711 species considered (see 'Evaluation' in Methods). By using our model outputs (see 'SDMs' in Methods) and phylogenetic and functional information, we calculated the multifaceted diversity distribution of the European Alps under several carbon emission and dispersal scenarios (see 'Diversity' and 'Uniqueness' in Methods) and compared these distributions between the present and the 2050 and 2080 horizons. We show that the European Alps are predicted to lose between ~7% and 16% of their total multifaceted diversity and uniqueness by 2080 (Extended Data Fig. 2 and Supplementary Fig. 1), and from ~19% to 27% if the flora would be unable to disperse (Supplementary Figs. 2 and 3). This loss is expected to occur primarily at low elevations with corresponding gains at higher elevations by 2050 and under the moderate SSP245 scenario (Fig. 1), primarily caused by large upward shifts of species distributions (Extended Data Fig. 3). Changes are even more exacerbated for 2080, the more severe SSP585 scenario and under unlimited dispersal (Supplementary Figs. 4–9).

Our results are in line with those of previous studies showing that most species are able to respond to climate change by migrating towards cooler temperatures[30–32], therefore increasing the short-term species richness of higher mountain strata[14,33,34]. On the one hand, by extending these results to species traits and evolutionary history, we show that upslope migrations infer a change not only in species richness but also in other biodiversity facets. On the other hand, these migrations also generally result in a decrease in species range size owing to limited physical habitat area, consequently explaining the

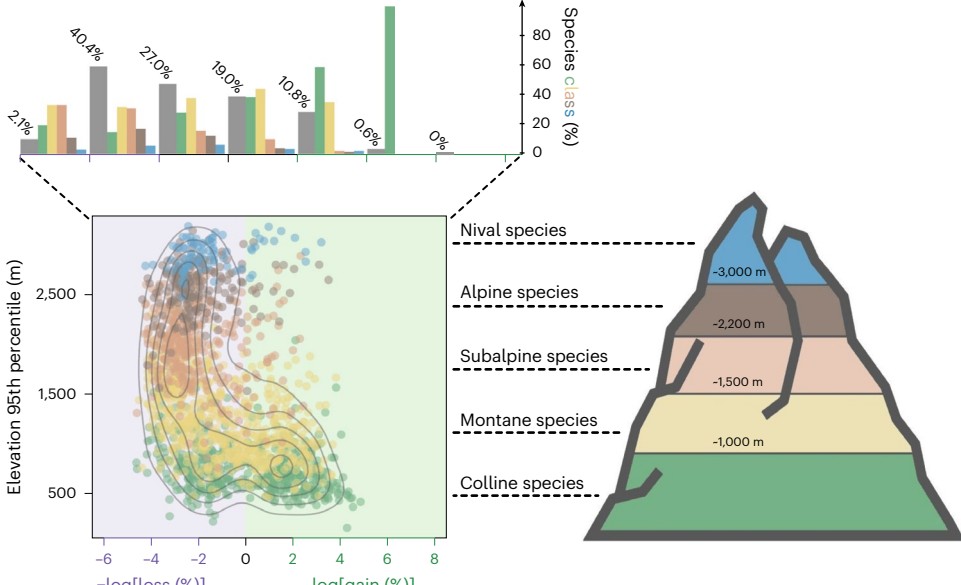

**Fig. 2 | Species range shifts by 2050 for SSP245, considering limited plant dispersal.** The scatter plot depicts for each species (points) its 95th percentile of extracted elevation values in function of its future range gain or loss (light green and purple background, respectively). In addition, FA elevation classes were assigned to each species (nival: blue, alpine: brown, subalpine: orange, montane: yellow, colline: green), and point density contour lines were drawn for further clarity. The upper bar plots summarize with a new *y*-axis and, for each gain and loss interval, the total proportion of species overall (numbered grey bar plots) and relative distribution of nival, alpine, subalpine, montane and colline species. Silhouettes from the Noun Project.

positive changes in multifaceted endemism and rarity that we uncovered at higher elevation[20,35].

In addition, we predict rural landscapes to be very species rich (Supplementary Fig. 10; for example, permanent crops and pasture) and to suffer from future land abandonments and forest successions over the study region (Supplementary Fig. 11), and land-use change to generally affect future biodiversity negatively (Supplementary Fig. 12). These results corroborate previous findings and further explain the future high loss of multifaceted plant diversity found in the European Alps. Rural (semi-managed) landscapes with a long land-use history (as in Europe) indeed harbour a high diversity of both species and habitats[16,36], especially grasslands[3,37]. As such, rural abandonment and land-use change in these regions create a loss of landscape heterogeneity, which normally benefit a wide range of organisms thanks to more resource opportunities[3,16,38], and generate detrimental effects on biodiversity[36]. Moreover, forest successions after abandonment have been reported to threaten alpine species, and their associated functions and physical habitats[39], potentially further describing positive changes of multifaceted endemism and rarity found in the Alps in the future.

### Species turnover and extinctions
For each species, model outputs (see 'SDMs' in Methods) were used to evaluate how much gain and loss in species range area are predicted to occur between current and future scenarios (see 'Post-analyses' in Methods). Overall, ~70% of species are predicted to lose areas of suitable conditions ('losers'), especially in higher mountain strata (Fig. 2), and to a greater extent by 2080 under SSP585 (Supplementary Figs. 13–15). The percentage of species losing the most of their suitable habitat increases from 2.1% by 2050 under SSP245 (Fig. 2) to 16.1% by 2080 for SSP585 (Supplementary Fig. 15). Among the remaining ~30%, many lowland species are instead forecast to experience strong range expansion ('winners'; Fig. 2) with larger gains by 2080 for SSP585 (Supplementary Figs. 13–15). The percentage of species expanding the most of their suitable habitat increases from 0.6% by 2050 for SSP245 (Fig. 2) to 4.2% by 2080 under the more severe SSP585 scenario (Supplementary Fig. 15).

On the one hand, the range losses illustrate loser species that are generally more restricted to specific environmental conditions and are forced to migrate upwards because of changing environments. As a consequence, not only does the species richness of higher mountain strata increase[14,33,34], but also, species lose distributions because of less available physical habitats and more physical barriers to dispersal leading to local population extinction ('dispersal lags')[15,20,40]. This also further explains the increasing multifaceted endemism and rarity at higher elevations. On the other hand, the range expansions illustrate winner species that are more able to adapt to novel environmental conditions (for example, thermophilic generalist species), inhabiting the lowlands, therefore conserving more range and expanding their distribution towards higher latitudes and elevations in the future[14,15,41,42]. This progressive species replacement across elevations is generally expected to increase over time and with increased global change[14,15]. Our results suggest that by the end of the twenty-first century, this climate-induced turnover will intensify, likely homogenizing the European Alps' plant communities (Supplementary Figs. 13–15) and possibly driving two plant species to extinction (*Antirrhinum latifolium* and *Iberis saxatilis*; Supplementary Table 1). However, as we did not account for competition between plant species in this study, we lack inclusion of important drivers of population dynamics and most probably underestimate potential extinction[43]. As such, some species are predicted to migrate from higher to lower elevations (Extended Data Fig. 3 and Supplementary Fig. 4), whereas they should instead be restricted from moving down the gradient because of high competitive exclusion from lower strata[5,15,44].

### Local conservation prioritization
In the face of important regional changes in climate and land use, comprehensive conservation planning that mutually emphasizes local and regional conservation prioritizations must be implemented[27]. Local prioritization (considering 'alpha' or pixel biodiversity) is a strategy that focuses more on protecting given localities and areas that are biodiversity and endemism rich within a given region, that is, local hotspots. For this, we used the additive benefit function (ABF) algorithm

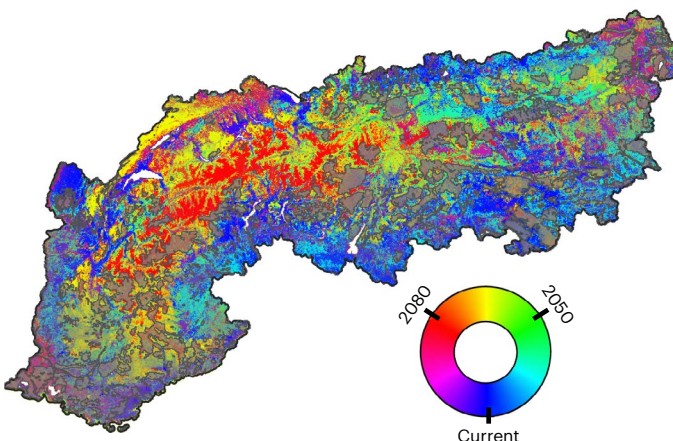

**Fig. 3 | Current and future conservation hotspots in the European Alps for SSP245, considering limited plant dispersal.** The colour scale represents areas that are higher priorities for current (blue), future (2050, green; 2080, red) or a combination of scenarios (for example, yellow is more important in the future relative to the present). They are determined on the method focusing on local biodiversity assemblages (Zonation ABF), and priority areas in grey with contours are those already protected under the current IUCN I and II and Emerald-Natura 2000 PA network.

of the Zonation software (see 'Zonation' in Methods) with our species model outputs and their phylogenetic and functional information. The primary objective is to evaluate how local conservation planning can potentially adjust to the various changes in biodiversity occurring in the European Alps over time and for different scenarios. Based on our projections, to effectively safeguard multifaceted diversity in the region (Extended Data Fig. 4), future conservation strategies should align with the anticipated upward shifts in plant multifaceted diversity and uniqueness projected under the moderate SSP245 scenario (Fig. 3) and the pessimistic SSP585 scenario (Supplementary Fig. 16).

Based on these results, a first clear common conservation strategy for the Alps involves assisting future plant upward migrations by increasing PA connectivity between elevation strata. In addition, we highlight areas of relative stability (overlapping priorities for expansion in the present and future) in the Mediterranean Alps (Fig. 4). The Mediterranean area is a biodiversity hotspot in Europe and is composed of distinct range-restricted and unique species[45,46]. Despite forecasted biodiversity loss, this region is still predicted to harbour high levels of multifaceted diversity and uniqueness in the future (Supplementary Figs. 17–20) that are still essential to protect. As most species (Extended Data Fig. 3 and Supplementary Fig. 4), lowland Mediterranean species were also detected to strongly expand their distributional range towards higher latitudes (Supplementary Fig. 21), which stresses the necessity of increasing reserve connectivity from south to north, and especially between the Mediterranean Alps and the central Alps. Species migration towards higher latitudes under global change is well documented[13,31,32] and is expected to increase as $CO_2$ emission rises over time (Extended Data Fig. 3 and Supplementary Fig. 4). Overall, we find similar conservation results when predicting conservation hotspots under simulations that did not initially include current PAs (Supplementary Figs. 22 and 23). This emphasizes that the current conservation network is insufficient to conserve biodiversity and requires adaptations to operate optimally under novel environmental conditions.

## Regional conservation prioritization

Unlike local optimization, regional conservation prioritization (maximizing 'gamma' or regional biodiversity) does not necessarily focus on single multifaceted-rich localities but rather on unique localities in

their composition and complementarity, that is, magnifying the multifaceted diversity of the whole region. For this, we used the 'core-area Zonation' (CAZ) algorithm of the Zonation software (see 'Zonation' in Methods) and evaluated how regional conservation strategies could adapt to multifaceted diversity changes in the European Alps and improve their protection. We forecast that predicted conservation hotspots at the regional level are similar to local ones (Supplementary Figs. 24 and 25), although more geographically distinct, and find that our regional conservation simulations protect multifaceted diversity as efficiently as local strategies (Extended Data Fig. 4).

While this corroborates the urgency to improve the connectivity of European PAs, this also stresses the imperative of better transnational conservation in the region. In line with the COP15 diversity framework (minimum of 30% land protection), we chose the top 20% of the network expansion for the current, 2050 and 2080 scenario. The conservation overlaps of the expansions expand the present conservation network from ~18% to ~35% of the European Alps' area, under SSP245 (Fig. 4) and SSP585 (Extended Data Fig. 5) scenarios. PAs are often biased towards higher elevation[18] and are generally known to be well adapted to species upward migration induced by global change[19]. On the one hand, we confirm this tendency as the current reserve network of the European Alps is predicted to better protect the species distribution of higher-elevation strata for future timelines and SSP scenarios (Supplementary Figs. 26–29). On the other hand, we detect that regional priorities are sporadically spread across the European Alps for both SSP scenarios (Fig. 4 and Extended Data Fig. 5). This leads to differences in the degree to which different countries should protect multifaceted diversity and calls for a more coherent and coordinated transnational reserve network across the European Alps.

As a result, the ideal contributions of France, Germany, Italy, Slovenia, Austria and Switzerland to such an optimized strategy differ strongly among elevation belts. To balance optimal transnational contributions, Switzerland would be expected to bear the largest efforts in expanding the network across all elevation strata owing to its very low PA coverage (~2% of the network; Supplementary Fig. 30). Austria would be expected to increase its PAs mostly at mid-elevation. France and Germany would be expected to redirect a higher focus on lowlands, where France could best contribute to a transnational strategy by focusing on PAs in the Mediterranean Alps (Fig. 4a). This latter statement is also valid for Italy and Slovenia who would best contribute to a complementary conservation network optimization by solely focusing on low to mid-elevations.

## European conservation perspectives

In the face of future multifaceted biodiversity extinctions and migrations, implementing conservation planning promoting diversity-rich areas (local conservation) and their irreplaceability (regional conservation) is a key strategy to fashion a resilient and adaptive conservation network in Europe. Here we show that adopting such a strategy exclusively in the European Alps should largely improve multifaceted biodiversity protection compared with the current sensu stricto (International Union for Conservation of Nature (IUCN) I and II plus Emerald-Natura 2000) or sensu lato (IUCN I–VI plus Emerald-Natura 2000) European conservation network[47] (Extended Data Fig. 4). These findings reaffirm previous studies indicating that the current European reserve network may not be well suited for effectively conserving biodiversity compared with our current understanding of its complete range and contributions to ecosystem functions[48–50]. As an example, the EU-Natura 2000 network was originally structured to protect habitats, rare species and migratory birds, thus omitting the multidimensional aspect of biodiversity[48].

Nevertheless, the EU-Natura 2000 framework is also the world's largest coordinated conservation network. Along with IUCN I and II categories, these PAs are often considered as the flagship tool of the EU reserve network owing to their benefits of protecting species diversity

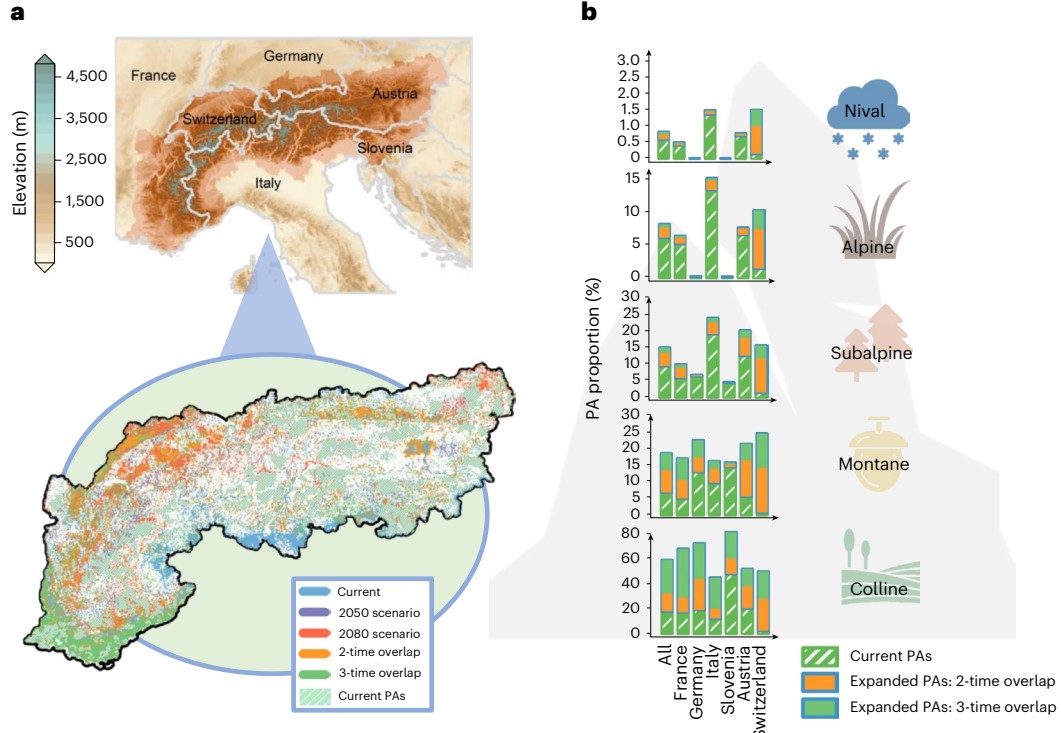

**Fig. 4 | Distribution of the current PA network of the European Alps and its future regional expansion for SSP245, considering limited plant dispersal. a**, The top panel depicts the geographic distribution of the Alps over France, Switzerland, Germany, Austria, Slovenia and Italy, while the bottom panel anticipates the conservation overlaps inferred from the top 20% expansion of the current sensu stricto network for each present, 2050 and 2080 scenario. Each SCP expansion was constructed using the CAZ algorithm (maximizing diversity of the whole region) and the reserve expansion approach of Zonation.

In total, the overlaps expand the network from ~18% (striped green; IUCN I and II and Emerald-Natura 2000) to ~35% of the study area. **b**, The conservation overlaps found in **a** are projected but distributed across national entities and elevation strata, with the *y*-axis describing how much the PA type (%) is predicted to be politically distributed when the network reaches 35% of land protection. Orange defines top expansion overlaps for two timelines, whereas green is for all timelines. Silhouettes from the Noun Project.

and rarity[48,49,51]. However, we also found that the convergences between the IUCN III and VI network and our expansion forecasts reach ~3% of the study area (Extended Data Fig. 6). Although more managed and primarily not regulated to protect species diversity[47,49], these PAs are already part of the current Alps' network and some of their multifaceted diversity levels are underestimated. We therefore suggest that future European reserve planning not only use IUCN I and II categories and the Emerald-Natura 2000 framework as conservation backbone, but also restore and readapt a targeted part of the IUCN III and VI network to more pristine protected areas. To that end, and as part of the EU Biodiversity Strategy for 2030, the EU should urgently adopt a novel directive coordinating the extension of its current sensu stricto network, to ensure its resilience to future environmental changes in protecting rich and irreplaceable areas of multifaceted diversity.

Ultimately, the success of future European planning will heavily rely on effective conservation coordination between both EU members and non-members. As part of the Emerald network, Switzerland is here a good example. Our study showed that the country is predicted to endure most of the effort in expanding the transnational network of the European Alps, owing to its very small amount of PAs that could not efficiently protect the unique and high multifaceted diversity of the region. Similar results were also found when considering IUCN I-VI categories within our SCPs (Supplementary Fig. 31), suggesting that if the conservation network of Switzerland was better integrated with the advanced EU network, the SCP objectives would be more balanced between the national entities of the region. As a major diversity hotspot in Europe, the Alps should be a central hub for the conservation planning strategy of the continent. Previous European and global studies

have already highlighted the importance of protecting this region[52-56]; however, Switzerland is in fact often missing from these priority assessments[49,50,57-61], despite its apparent crucial geographic and diversity importance in European conservation. Overall, biodiversity knows no political borders and increasing local conservation and connectivity at the countries' edges will be a necessity to ensure safe corridors between ecoregions (for example, Mediterranean to central Alps) and assist species in their latitudinal and elevation migrations.

## Challenges

The establishment of a novel European reserve network will also face data availability challenges, and future studies assisting national entities in this task will need to solve these limitations that conservation assessments are currently facing.

First, this study did not consider the future dispersal of vegetation from outside our study extent. This was mainly because of observation and dispersal data limitations, as no such extensive dataset of precise ecological information exists over Europe yet. While studying a larger extent and number of species would have allowed the future lowland diversity of the region to be better assessed, we also want to highlight that due to the absence of biotic interactions in our models, the outcome of how outside species migrations would impact (positively or negatively) the future diversity of the Alps is hard to predict. Overall, further efforts have to be made to agglomerate enough ecological data for many taxa to achieve comprehensive conservation planning in Europe. The recent exponential growth of ecological data repositories and opportunistic observations[62-64] will certainly help in this task, but more European coordination in retrieving and achieving harmonized

datasets is necessary. Moreover, we here focused on plants because, unlike other taxonomic groups, the information on vegetation dispersal modes and rates is increasingly available[65–67], which, combined with our observational dataset, allowed the scope of this study. Dispersal data and proxies for many taxa are needed if such dynamic processes are to be included in future European conservation planning. This also echoes the 'iniMatAge' parameter used in our semi-mechanistic dispersal simulations (species' initial maturity to disperse; Supplementary Table 7). No information was available in the literature, and a default parameter of 2 years was set to all species, therefore increasing the uncertainty of our results on how fast the predicted changes of biodiversity will occur and how quick the conservation planning recommended here should be applied. Ideally, and as a prevention, the future European network therefore needs to implement strong elevation and latitudinal PA connections by 2030 to rapidly assist species in their migration.

Second, recent developments have highlighted several perspectives of concomitant conservation prioritization of species diversity and NCPs[60,68,69]. Although biodiversity highly contributes to ecosystem functioning and services[70], their conservation demands may differ[71,72]. Multifaceted diversity, and regulatory (for example, pollination and carbon sequestration) and (non-)material NCPs (for example, water, ecotourism and heritage landscapes), should therefore be considered together within one common transnational reserve network, ensuring the maximization and protection of the multifaceted diversity and NCPs of Europe. Future studies should integrate SCPs that include such ecological features together to ensure efficient conservation planning in Europe. Finally, biodiversity, although driven by species and their evolutionary history, functional roles and abundance[21,73], is also tightly linked to biotic interaction, both being mutually dependent on each other[74–81]. Previous literature has stressed the need of conserving these interactions to ensure a better protection of diversity and nature services[82–85]. Including biotic interactions in SDMs to finely explain species distribution, and for better conservation planning, faces numerous challenges[86–90], and novel approaches are needed to disentangle their importance in shaping biodiversity patterns. Empirically assessing food webs is increasingly done over macroecological scales[81,91–93]. Therefore, spatially informing these networks over Europe and including them within SCPs, alongside multifaceted diversity and NCPs, would improve our understanding of their mutual correlation link and allow a more comprehensive and adaptive European reserve network to be obtained.

In the face of these challenges, there is therefore an urgent need to implement a comprehensive conservation planning of multifaceted diversity and uniqueness over the whole continent. The current European network must be redefined into one complementary and resilient transnational conservation framework, which will efficiently protect the whole biodiversity of the European flora and fauna at present and for future environmental changes.

## Methods

### Study area and observations
**Study area.** The study area covered the European Alps, as defined by an enlarged version of the official Alpine Convention perimeter[94]. The enlargement consisted of adding Switzerland entirely, as well as two French departments, that is, Ain and Bouches-du-Rhône, for which we had well-documented species observations. In addition, we extracted the IUCN category I and II and Emerald-Natura 2000 reserve network of the study area from the World Database on Protected Areas (https://www.protectedplanet.net/en).

**Observations.** The final observational dataset used in this study included (1) a compiled dataset from 75 various sources (~71%) and (2) a large compilation of observations extracted from the Global Biodiversity Information Facility (GBIF; http://www.gbif.org/)[95,96]. While

(2) is available on the EnviDat repository (https://doi.org/10.16904/envidat.371) or GBIF (https://doi.org/10.15468/dd.mb6jzt), (1) can be shared only upon reasonable requests owing to the various data policies and sensitive locations of rare species (see Supplementary Methods 1 for more details).

In total, our observational dataset included 6,655,163 unique observations accurate to 11.1 m for 4,250 species (Supplementary Table 2 and Supplementary Fig. 32a) and ~29% of records from GBIF. This set was further filtered according to the prevalence of each species (or proportion of 100 m pixels occupied); that is, species occurring in less than 30 pixels across the study area were removed. In total, the refined observational dataset included 3,167 species used in model calibration (see Supplementary Fig. 32b and Supplementary Table 3 for further description). It is important to note that for species with >10,000 observations, we sampled randomly without replacement a subset of 10,000 observations for better computation efficiency[97,98]. In addition, an independent and unbiased test dataset, reporting the empirical and distributional range of our 3,167 plant species over the European Alps, was constructed from expert-based information available in the *Flora Alpina* (FA)[99] and the extraction of the 5th–95th percentile elevation values of each species (see Supplementary Fig. 33 for more information).

### Environmental data
**Climate.** Climate information was extracted from the Climatologies at High Resolution for the Earth's Land Surface Areas (CHELSA v2.1) portal[100] (https://chelsa-climate.org/) and more specifically using the new 'chelsa-cmip6' Python library[101]. This library allows, for any available climate models and time periods (https://esgf-node.llnl.gov/search/cmip6/), novel CHELSA outputs to be automatically generated for a specific geographical extent. In total, four bioclimatic predictors, known to have major ecophysiological effects on plant life[19,102,103], were extracted for the European Alps extent: growing degree days (GDD), annual precipitation (BIO12), temperature (BIO4) and precipitation seasonality (BIO15). These predictors were obtained at 1 km resolution, for current (time period 1981–2010) and future climate (2041–2060 and 2071–2090), for 14 CMIP6 scenarios, that is, 2 SSP emissions (SSP245, SSP585)—an updated equivalent of the representative concentration pathways 4.5 and 8.5, respectively—and 7 global climate model scenarios (GFDL-ESM4, MIROC6, AWI-CM-1-1-MR, EC-Earth3, IPSL-CM6A-LR, INM-CM5-0, MPI-ESM1-2-LR).

**Soil.** We derived soil property layers at a 100 m resolution over the study area by mapping ecological indicator values (EIVs)[104–107] in space following the method described in a previous study[105]. First, we obtained plant EIVs from FA[99] and retained two different EIVs to characterize the local edaphic conditions: soil nitrogen (EIV-N) and soil substrate composition (EIV-G). Based on the plant EIVs of our 4,250 species extracted from FA and all related observations (~6,655,163 records), indicator value maps for soil nitrogen and substrate composition were extrapolated using random forest as described in Supplementary Methods 2. All generated EIV soil property layers showed excellent evaluations with Spearman $r > 0.82$ (Supplementary Table 5). The generated EIV soil property layers are proxies of soil nitrogen (NITROGEN) and substrate composition (CALCAREOUS%), and have been shown to be excellent predictors of plant species distribution in SDMs[105,108]. It is important to note that given the unavailability of future predicted soil information, we considered current and future soil unchanged. See Supplementary Methods 2 for full details on this section.

**Land cover.** LC is also known to have a strong influence on species distributions[5,109–111]. Therefore, two LC-change projections were obtained from the EU-funded ALARM–ECOCHANGE and VOLANTES–HERCULES projects[112–115] at 1 km resolution, each including 6 original (grassland, forest, built-up, cropland, permanent crops, others) and 10 reclassified

(pasture, semi-natural vegetation, forest, built-up, permanent crops, irrigated and non-irrigated arable land, recently abandoned pasture and arable land, others; Supplementary Table 6) LC categories, respectively. While the current LC is derived from the CORINE 2000 classification[116], each LC-change projection included two emission scenarios consistent with SSP245 and SSP585 (refs. [117,118]), namely, B1-SEDG (Sustainable European Development Goal) and A2-BAMBU (Business-As-Might-Be-Usual). The four LC scenarios were available for the time period 2041–2060 (for ECOCHANGE and HERCULES) and 2071–2090 (for ECOCHANGE only). It is important to note that, while ECOCHANGE provided future LC projection for the whole European Alps, the original HERCULES outputs did not include Switzerland. Therefore, using the same methodology[114,115], new HERCULES LC projection scenarios including the whole of Switzerland were generated.

**Correlation.** All predictors were projected to the standard Lambert azimuthal equal area projection for Europe (EPSG:3035), and continuous current predictors (climate and soil) showed Pearson's inter-correlation $|r| < 0.7$ (Supplementary Fig. 34), as suggested when model projections outside the calibration range are involved[103].

### Observer bias correction

**Bias covariate correction.** Our observational dataset originated from a range of different sources that often lack sampling design; therefore, a strong geographic bias towards Switzerland and France was present in our refined observational dataset (Supplementary Fig. 32). To correct for this bias, three potential bias covariates were generated over the study area[98,119]: (1) the target group observation density and distances to (2) roads and (3) cities. Observation density, which included all species records of our original observational dataset (6,655,163 observations for 4,250 species), was calculated by sum aggregation to a 100 m grid (Supplementary Fig. 32a), which allowed a general observer bias to be defined across our study area[98]. Distances to roads and cities were generated based on OpenStreetMap (https://www.openstreetmap.org). All roads and cities of the study region were extracted from this source and converted into two binary 100 m grids. Distances to roads and cities were then independently calculated with GDAL/OGR 3.8.0 and Python 3.9 (function 'gdal.ComputeProximity', https://gdal.org/). Along the environmental predictors, our three bias covariates were then used within each SDM to fit the species observation to a potential sampling bias when detected. All bias covariates were projected to EPSG:3035, after square root transformation[119,120]. It is important to note that all bias covariates were weakly correlated with climate and soil, that is, Pearson's $|r| < 0.3$ (Supplementary Fig. 34). Environmental effects were therefore hardly masked by observer-bias effects during model calibration[120].

**Environmental bias correction.** Before data collection, the appropriate sampling design should be environmentally stratified[121–124]. Sampling frequencies in environmental space may in fact still remain skewed if species observations are not initially sampled according to an environmental stratification. Therefore, to further address the environmental bias in the sampling design of our refined observational dataset (Supplementary Fig. 35), a recent corrective method, based on environmental stratified resampling of the observational dataset, was implemented before model calibration using the R function wsl.ebc[98]. Environmental bias correction (EBC) corrects potential environmental bias in the design of an observational dataset, by artificially subsampling original species observations based on a chosen number of environmental clusters over the study area[125]. In total, EBC was applied to only 1,248 species whose observations were detected to be environmentally biased. The resulting corrected observations and their environmental frequencies (before and after EBC) may be found in Supplementary Fig. 36. See Supplementary Methods 3 for full details on this section.

### SDMs

**Calibration.** For each species, model calibrations were done at 100 m resolution, by including current climate (1km), LC (1 km), soil (100 m) and our bias predictors (100 m), and were done twice, that is, one model per categorical LC. Along with an elastic net regularization[126,127], we used a special case of presence-only SDM, namely, point-process models (PPMs), whose output represents the intensity of the expected number of species occurrences per unit area, which is modelled as a log-linear function of the environmental covariates[119,128,129]. Although described as an equivalent of MAXENT[130], PPMs have many more methodological benefits[98,119,128].

First, unlike most SDM approaches[131,132], PPMs propose an automated framework to choose the adequate number and location of 'quadrature points' (commonly referred to as 'background points' or 'pseudo-absences')[119] if no true absences are available. Second, on top of dealing with observer bias more objectively[98] (see 'Bias Covariate Correction' in Methods), PPM indirectly avoids incomplete species response curves by randomly sampling quadrature points across the whole environmental gradient[119,128]. Finally, PPMs may be easily used with lasso and clarifies the form of the modelled response as it represents an intensity of species observation (or abundance) and not a probability[119,130]. See Supplementary Methods 4 for full details on this section.

**Evaluation.** We evaluated the predictive performance of each PPM against the FA test dataset by using five-fold spatial block split-sampling tests[133]. This approach involves preliminarily delineating independent spatial blocks to partition observations in geographic space. Here, for each species, we evenly partitioned its observations, quadrature points and FA presences and absences into 10 blocks and combined them to 5 folds (see Extended Data Fig. 7 for more details). PPM performance was evaluated using FA presences and absences of the left-out fold, the true skill statistics (TSS) and the Boyce index. While TSS evaluates matches and mismatches between binary observations (here FA presences and absences) and model predictions[134], the Boyce index is a presence-only metric that measures the expected predicted-to-expected ratio of presences (here FA presences) in each class of predicted values[135,136]. Both TSS and Boyce index range from −1 to +1, and models performing poorly—that is, concurrently having a TSS and Boyce index <0.3—were removed.

### Projection

**Unlimited dispersal.** For each species, retained calibrated models were projected to 100 × 100 m resolution over the study area for the current (time period 1981–2010) and future (2041–2060 and 2071–2090) environment of our 14 CMIP6 and 4 LC scenarios, by setting the 3 bias covariates to a constant value of 0 for all cells to correct for the fitted observer bias[98,120] (also done for evaluation; see Extended Data Fig. 1 for the method workflow summary). Obtained intensity projections (or abundances) were then averaged across all CMIP6–LC scenarios to generate per species one current (2000) and four future (2050-SSP245, 2050-SSP585, 2080-SSP245 and 2080-SSP585 ensembles) SDM intensity maps. Such abundance maps do not include values strictly equal to zero, which are essential to infer species range gains and losses, and risk assessments. Therefore, all SDM intensity maps were also converted to SDM presence and absence maps using the 'maximum TSS' average of each species (maxTSS mean of the retained calibrated models) and intersected together to generate SDM intensity-and-absence maps.

**Limited dispersal.** When using SDMs, one major inconvenience is to account for species dispersal limitation. Standard SDM projections in future environmental conditions implicitly assume unlimited dispersal. Said differently, model predictions of future changes in the distribution of a species indirectly presume that the species can colonize any suitable environmental habitats or pixels regardless of its location. This is a

problem as many geographic (physical barriers such as rivers, forests or mountains) and ecological (species dispersal capacity) features could impede the species from dispersing too far from its initial distribution. We therefore included in our future SDM intensity-and-absence maps the mechanistic process of dispersal by using the R package 'MigClim' (function 'MigClim.migrate')[137,138], which operates based on the current and future binary distribution of a given species and its true ecological information on dispersal distances in metres (see Supplementary Table 7 for details on all parameters and data)[139–145]. Yearly maximum and minimum dispersal (in metres) was extracted from the literature[65,66] and available for 1,711 species (compiled data available at https://doi.org/10.16904/envidat.371). Using these data, along with our current and future SDM presence-and-absence maps, we generated for each species four future MigClim binary maps accounting for limited dispersal. Finally, to convert back future MigClim binary outputs to intensity-and-absence maps, the former was intersected with the original SDM intensity maps of each future scenario.

**No dispersal.** Finally, the last four future SDM intensity-and-absence maps considering no dispersal were generated following a previous study[146]. To that end, the four (unlimited dispersal) SDM intensity-and-absence maps of each species were intersected with their current SDM presence-and-absence distribution; that is, we kept as suitable areas only those concurrently occurring for present and future conditions.

Finally, all final layers (that is, 13 per species) were each aggregated by mean from $100 \times 100$ m to $1 \times 1$ km resolution, as stacked SDMs provide more meaningful predictions of species diversity when species distributions are aggregated from high to lower resolution[19,108,147].

## Multifaceted diversity and uniqueness

**Diversity.** For current and each of the 12 dispersal scenarios, spatial taxonomic, phylogenetic and functional diversity were calculated as abundance-based diversity with Hill numbers and their recent extensions[73,148–150]. To that end, we used the aggregated species distributions, the R package 'V.PhyloMaker'[151–153], a constructed functional tree based on four plant trait values (mean plant height, leaf dry matter content, specific leaf area and leaf carbon-to-nitrogen ratio)[154–165] and the R package 'hillR'[73,166]. Here we chose for all metrics Hill order ($q$) = 1, that is, an average sensitivity of diversity to species abundance (or occurrence intensities). It is generally known that phylogenetic diversity (PD) and functional diversity (FD) are not independent from taxonomic diversity (TD)[21,167]. The residuals of two linear regressions of TD on PD and FD (quadratic terms included) were therefore extracted to generate new layers of relative phylogenetic and functional diversity (rPD and rFD, respectively)[168–170]. See Supplementary Methods 5 for more details.

**Uniqueness.** Using the same data resources as above, we calculated for each scenario the weighted taxonomic[25] (WE), phylogenetic[26] (PE) and functional endemism (FE)[27] across the study area with the R package 'phyloregion'[171]. Relative phylogenetic and functional endemism (rPE and rFE, respectively) were generated following the same procedure and justifications as for rPD and rFD. Phylogenetic[172] (PR) and functional rarity[28] (FR) were also calculated using the R package 'funrar'[173]. See Supplementary Methods 6 for more details.

## SCP

**Zonation.** Conservation prioritizations were run using the conservation planning software Zonation 4.0 (refs. 174,175). Zonation ranks cells of a considered region from lowest (0) to highest (1) conservation values, based on both the irreplaceability and complementarity of input ecological features[60,90]. Zonation computes the conservation values of all conservation units (in this case, raster cells) based on the distribution of all features, and iteratively removes cells with the lowest conservation values until all are removed[176].

Unlike Marxan[177] or Prioritzr[178], Zonation is not intended for target-based conservation planning[175,179,180], defined as protecting the distribution of each biodiversity feature of interest (for example, species, habitat, ecosystem services) up to a specific percentage (user configurable) at minimum cost. Instead, Zonation is most useful and efficient when a very large set of biodiversity features is available, and provides at once ranked priority areas over the whole study region as outputs[175,179]. To that end, how much each feature should be protected is therefore not predefined by the user but an emergent property of the prioritization process[181]. This allows Zonation to better implement a complementary protection of biodiversity hotspots over the landscape by prioritizing clear irreplaceable areas with unique feature assemblages[175,179,181]. Finally, we chose Zonation because of its computational efficiency, ability to process very large rasters at high resolution without any memory issue, and permission of both binary and continuous biodiversity features as input[175,179–182].

For each current and dispersal outcome, we ran multifaceted-based prioritizations to maximize concurrently the representation of species, and phylogenetic and functional distinctiveness. For this, our aggregated species distributions were used as Zonation features, and each species layer was weighted by the sum of its phylogenetic and functional uniqueness (see 'Uniqueness' in Methods). While other SCP studies have accounted for multifaceted diversity by using different methods[27,54,183], we decided to use this novel and more intuitive approach so that, on the one the hand, phylogenetic and functional diversity and uniqueness aspects were concurrently included and, on the other hand, only one SCP map per current and future scenario was kept. This decision allowed us to better integrate all facets of biodiversity under one conservation roof, while analysing each scenario separately to evaluate the individual solution generated by each timeline, $CO_2$ emission and dispersal type, and to determine how much they diverge. Other methods, such as additional prioritization runs on phylogenetic and functional branch distribution[27,182], would have drastically increased the amount of SCP outputs, impeding conservation planning clarity.

In total, we ran 52 prioritization scenarios (see Supplementary Table 9 for parameters), that is, for each current and dispersal scenario outcome ($n$ = 13), accounting for two prioritization allocation approaches ('optimal reserve selection' and 'reserve network expansion') and using the CAZ and ABF prioritization algorithms. The 'selection' approach identifies the highest-priority areas on the entire landscape without accounting for the current configuration of PAs. The 'expansion' approach considers the current reserve network of the Alps including PAs designated as IUCN I and II and Emerald-Natura 2000 categories. In this approach, areas outside of PAs are ranked allowing for identification of the highest priorities outside the current PA network that best complement protected biodiversity. For the main regional and transnational conservation strategies (Fig. 4), the 'expansion' approach was used together with the CAZ algorithm, which assigns conservation values by maximizing regional diversity and its complementary, that is, by minimizing the extinction of features and protecting the worst-off ones (those with very little distribution remaining). For the main local and national conservation strategies (Fig. 3), the 'expansion' approach was used together with the ABF algorithm, which assigns conservation values by maximizing local diversity hotspots and their complementarity, that is, by minimizing the extinction of local multifaceted richness. In total, 52 SCP maps were generated (see Supplementary Table 10 for a summary).

**Post-analyses.** In line with the recent COP15 biodiversity framework of the Convention on Biological Diversity, we chose the top 20% of the current, 2050 and 2080 'reserve network expansion' simulations (Fig. 4), to correctly project a conservation overlap that would extend the present PA network to ~30% of the European Alps' surface. For each percentage

of expanding PAs over the study region, we calculated the cumulative representation of species, phylogenetic and functional branch occurrence intensities[184], of their range and of species functional and phylogenetic rarity (Extended Data Fig. 2 and see Supplementary Methods 7 for more details). In addition, we defined 'expanded PAs' as the 2- and 3-time overlap of the top 20% reserve expansion of current, 2050 and 2080 prioritizations (Fig. 4). The 95th elevation percentile was calculated per species by extracting the values of a digital elevation model with species observations (Fig. 2 and Supplementary Fig. 33). Finally, the percentage of gains and losses (Fig. 2) was calculated for each species based on their range, that is, the number of pixels of the study region that the species occupies, as follows:

$$\frac{|\text{Range}_{\text{Future}} - \text{Range}_{\text{Current}}|}{\text{Range}_{\text{Current}}} \times 100 \qquad (1)$$

## Reporting summary

Further information on research design is available in the Nature Portfolio Reporting Summary linked to this article.

## Data availability

All data, code and materials supporting the findings of this study are available in the EnviDat repository (https://doi.org/10.16904/envidat.371), which provides options to download single (via WGET and singular FTP links) or all files (S3 access using the software Cyberduck or any other S3 clients) used in this project.

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

## Acknowledgements

We thank P. Descombes for his help in generating the soil predictors; L. Boulangeat for the trait measurements; J. Renaud for his help in cleaning the species data and taxonomy; M. Guéguen for providing the downscaled climate predictors; S. Dullinger, A. Guisan, J. Alexander, R. Marsh and L. Roux for fruitful discussions; and all of the various contributors for sharing numerous datasets of species observations and traits, and for having made possible the large-scale scope of this study. We particularly thank Info Flora and the Conservatoire Botanique National Alpin for their continuous effort in collecting and cleaning plant data across Switzerland and the French Alps. This work was supported by the ANR-SNF bilateral project OriginAlps, with grant numbers 310030L_170059 (Y.C.-M., N.E.Z.) and ANR-16-CE93-004 (W.T., S.L.). W.T. also acknowledges the French Biodiversity Office and the Region Sud for financial support and the HorizonEurope NaturaConnect project (no. 101060429).

## Author contributions

Conceptualization: Y.C.-M., N.E.Z. and W.T.; methodology: Y.C.-M., L.J.P. and W.T.; software: Y.C.-M., P.H.V. and D.N.K.; investigation: Y.C.-M.; visualization: Y.C.-M.; formal analysis: Y.C.-M.; funding acquisition: N.E.Z., S.L. and W.T.; supervision: N.E.Z. and W.T.; writing—original draft: Y.C.-M.; writing—review and editing: Y.C.-M., L.J.P., P.H.V., D.N.K., L.P., S.L., N.E.Z. and W.T.

## Funding

## Competing interests

The authors declare no competing interests.

## Additional information

**Extended data** is available for this paper at https://doi.org/10.1038/s41559-023-02287-3.

**Correspondence and requests for materials** should be addressed to Yohann Chauvier-Mendes.

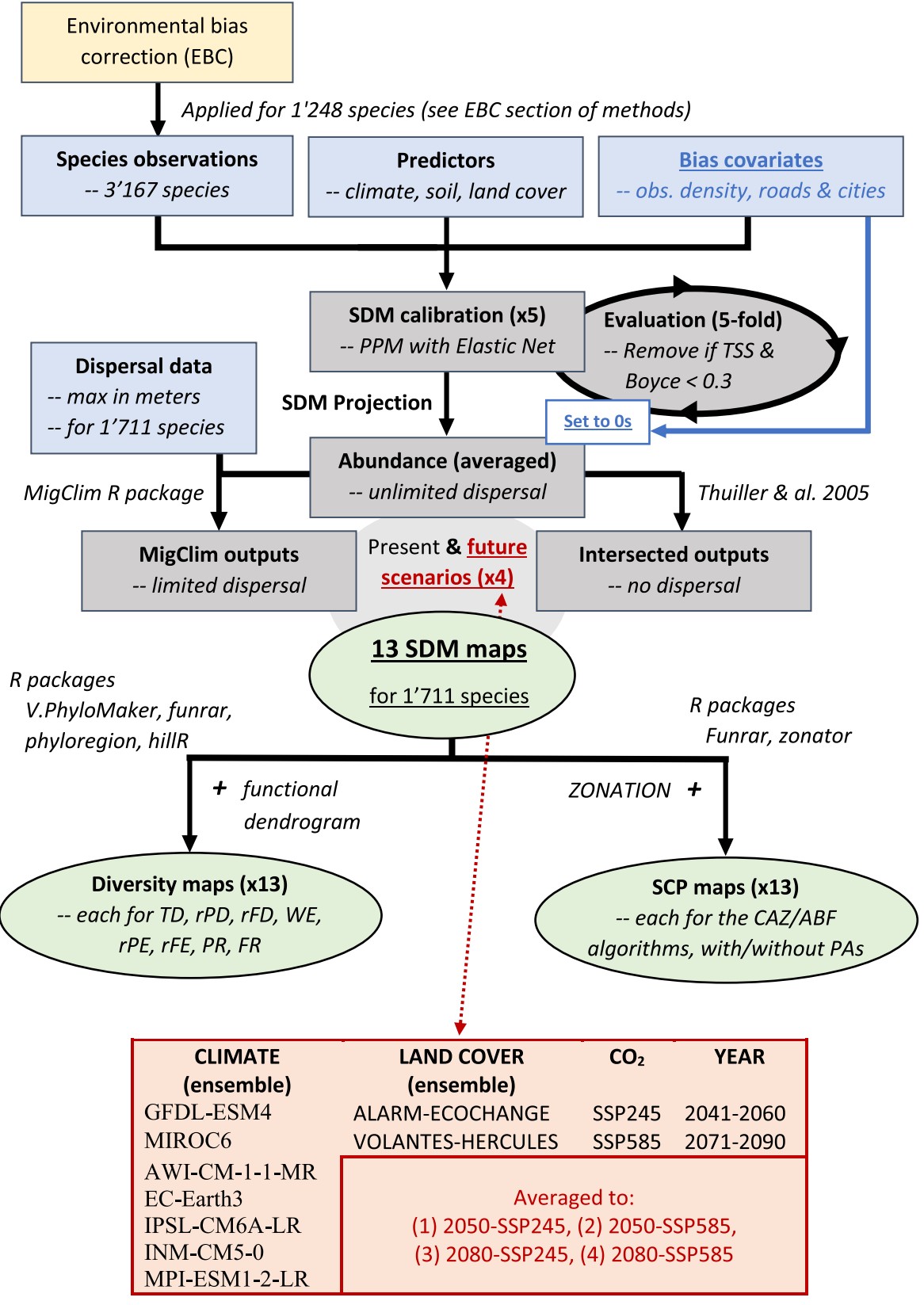

**Extended Data Fig. 1 | Methods workflow.** It is important to note that, overall, 42 future scenarios were employed to project and average our SDMs, since VOLANTES-HERCULES was available only for the year 2041-2060. Blue rectangles specify the input data, while green rectangles specify the model outputs and their derivatives. TSS: True Skill Statistics, SCP: Systematic Conservation planning, CAZ: Core-Area Zonation, ABF: Additive Benefit Function, TD: Taxonomic Diversity, rPD: relative Phylogenetic Diversity, rFD: relative Functional Diversity, WE: Weighted Endemism, rPE: relative Phylogenetic Endemism, rFE: relative Functional Endemism, PR: Phylogenetic rarity, FR: Functional Rarity. Silhouettes from the Noun Project.

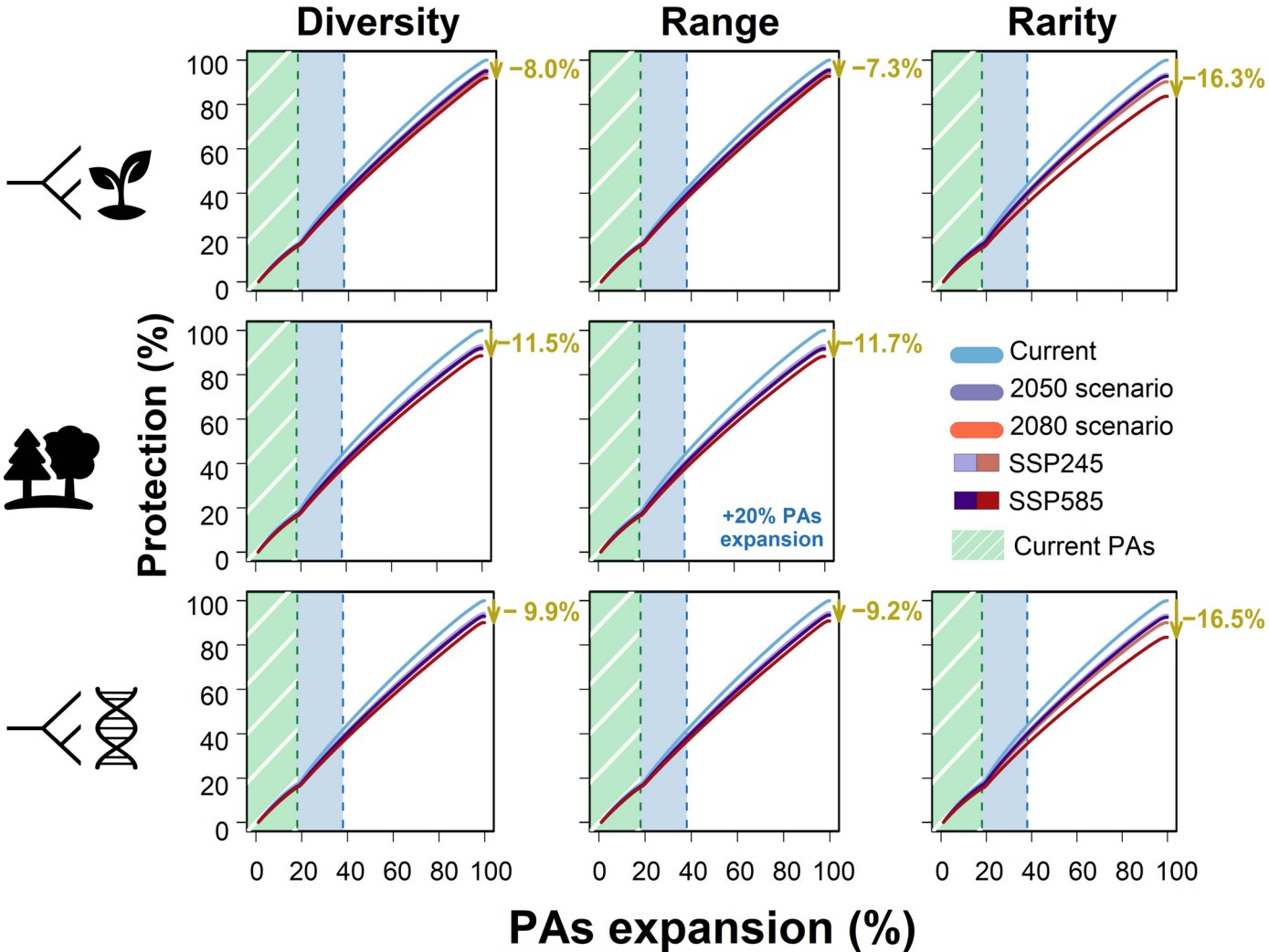

**Extended Data Fig. 2 | Reserve network cumulative expansion for *Zonation ABF* and limited plant dispersal.** First, second and third row depict the functional, taxonomic, and phylogenetic dimension, respectively. Panels describe the cumulated protection of plant multifaceted diversity and uniqueness in function of the reserve network expansion (% of the study region) for each current and future conservation simulation. Yellow percentages display the amount of feature loss between the current and 2080 prioritisation. Current PAs of the European Alps cover ~18% of the study region and are displayed in striped green. Results of expanding the reserve network by 20% are displayed over the panels in blue. It is important to notice that there is a strong expansion overlap between 2050-SSP245 and 2050-SSP585, hence the difficulty to sometimes decipher the five coloured curves. Silhouettes from the Noun Project.

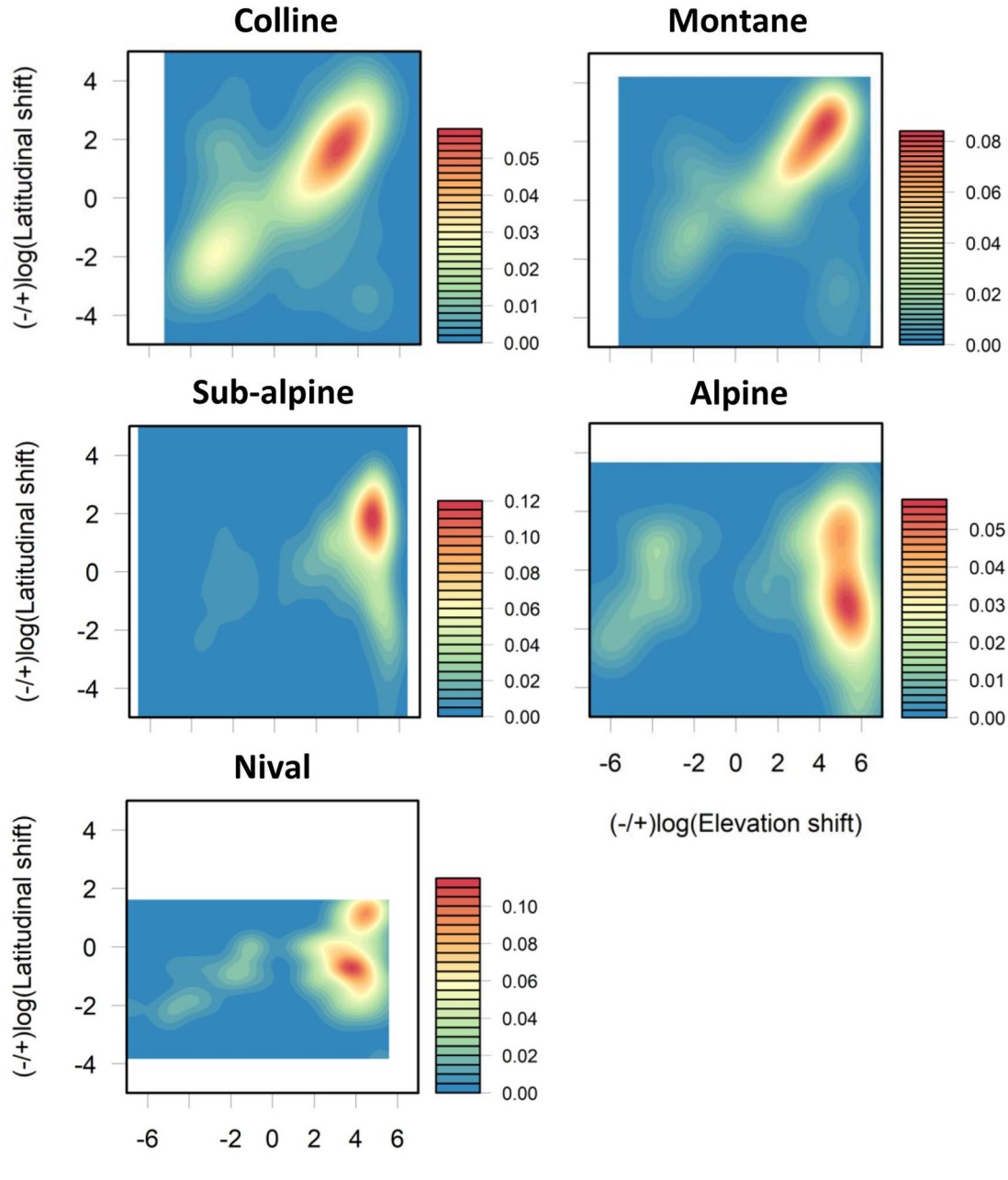

**Extended Data Fig. 3 | Latitudinal and elevation changes in species ranges by 2050 for SSP245 and limited plant dispersal.** Heat maps represent, for each species elevation class, the density of its predicted spatial changes in a two-dimensional space. For each species, latitudinal shifts were calculated by subtracting the median latitude of its current range (Y coordinate of each 1×1-km pixels) to that of its future range, whereas elevation shifts were calculated by subtracting the median elevation of its current range (values extracted from DEM over Europe - EU-DEM; https://www.eea.europa.eu/data-and-maps/data/eu-dem) to that of its future range. All species data points were then plotted and summarized as a density heat maps.

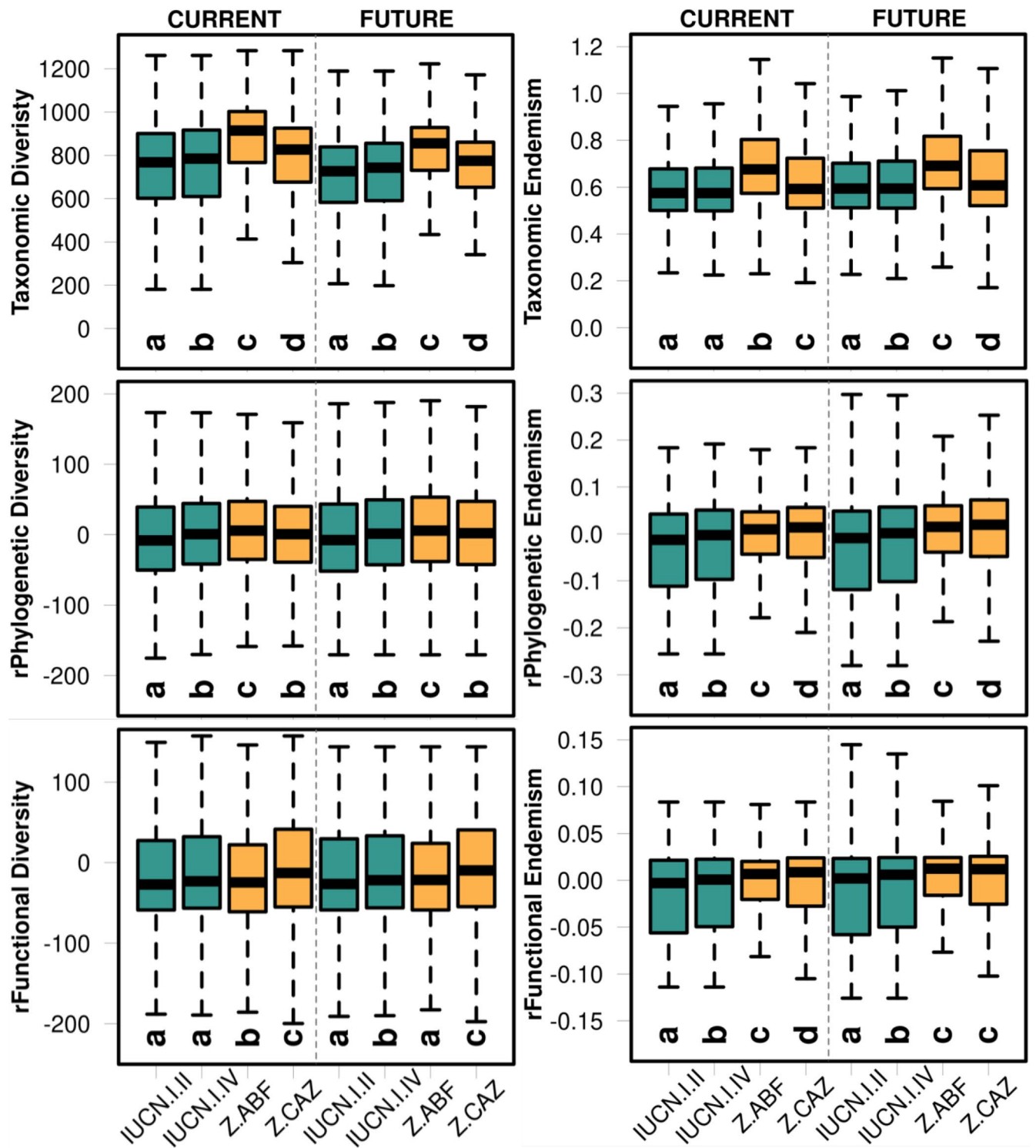

**Extended Data Fig. 4 | Multifaceted diversity GAP analyses between different types of reserve network for current and one future scenario (2050, SSP245 and limited plant dispersal).** Left 'current' panels depict the level of present protection of each network for taxonomic, weighted endemism, relative phylogenetic, functional diversity, and endemism, whereas right 'future' panels depict the level of future protection. *IUCN.I.II* includes IUCN categories I-II and Emerald-Natura 2000; *IUCN.I.VI* includes IUCN categories I-VI and Emerald-Natura 2000; *Z.ABF* includes IUCN categories I-II, Emerald-Natura 2000 and ABF expanded PAs (overlaps of the top 20% current and future conservation

scenarios generated with the ABF *Zonation* algorithm); *Z.CAZ* includes IUCN categories I-II, Emerald-Natura 2000 and CAZ expanded PAs. Kruskal-Wallis tests were here applied for each panel (***$p$-value ~ 0 for all). All pairwise comparisons (two-sided) were run with post-hoc Dunn tests, Bonferroni correction (adjustment for multiple comparisons) and displayed following a letter-based representation (*$p$-value < .05). For each panel, n = 361'300 cells examined over four independent reserve network types. Boxplots indicate median (middle line), 25th and 75th percentile (box), and minimum and maximum (whiskers).

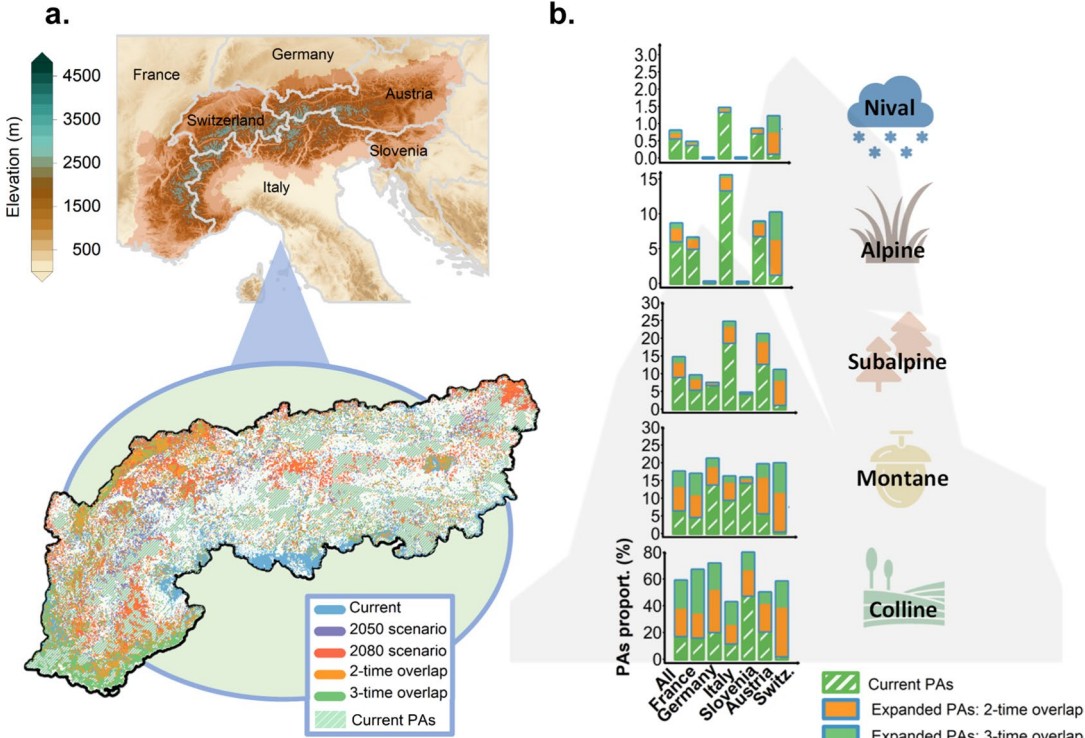

**Extended Data Fig. 5 | Distribution of the current protected areas (PAs) network of the European Alps and its future regional expansion for SSP585 and limited plant dispersal.** Top panel depicts the geographic distribution of the Alps over France, Switzerland, Germany, Austria, Slovenia and Italy, while bottom panel anticipates the conservation overlaps inferred from the top 20% expansion of the current *sensu stricto* network for each present, 2050 and 2080 scenario. Each SCP expansions was constructed using the CAZ algorithm (maximizing diversity of the whole region) and the reserve expansion approach of *Zonation*. In total, the overlaps expand the network from ~18% (striped green; IUCN I-II and Emerald-Natura 2000) to ~35% of the study area. (**b**) projects the conservation overlaps found in (**a**) but distributed across national entities and elevation strata, with the Y axis describing how much PAs type (%) is predicted to be politically distributed when the network reaches 35% of land protection. Orange defines top expansion overlaps for two timelines, whereas green is for all timelines. Silhouettes from the Noun Project.

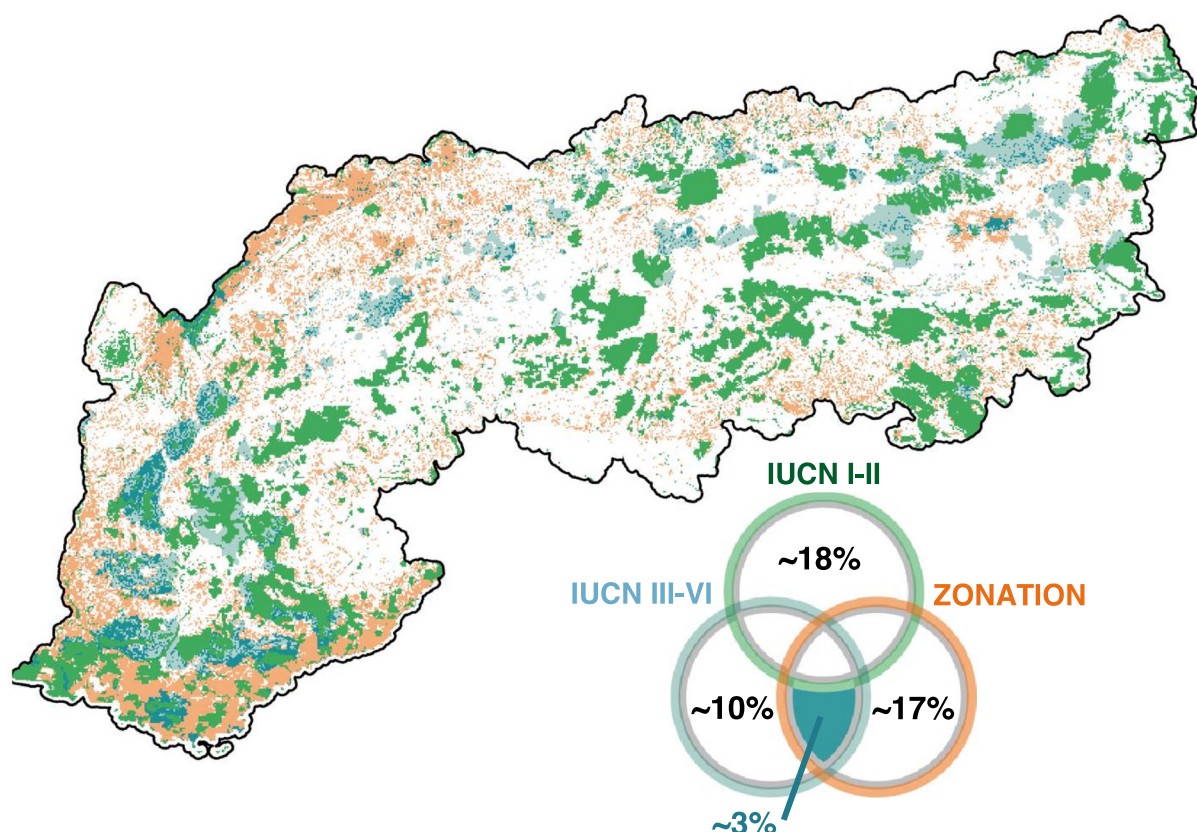

**Extended Data Fig. 6 | Distribution of areas of conservation importance in the European Alps for SSP245, using *Zonation* CAZ and limited dispersal.** Colour green, light blue and orange depict the current distribution of IUCN categories I-II, III-VI and *Zonation* expansion forecasts respectively. Percentages are expressed relative to the area of the study region. Convergences between IUCN III-VI and the *Zonation* forecasts were found to reach 3% of the European Alps.

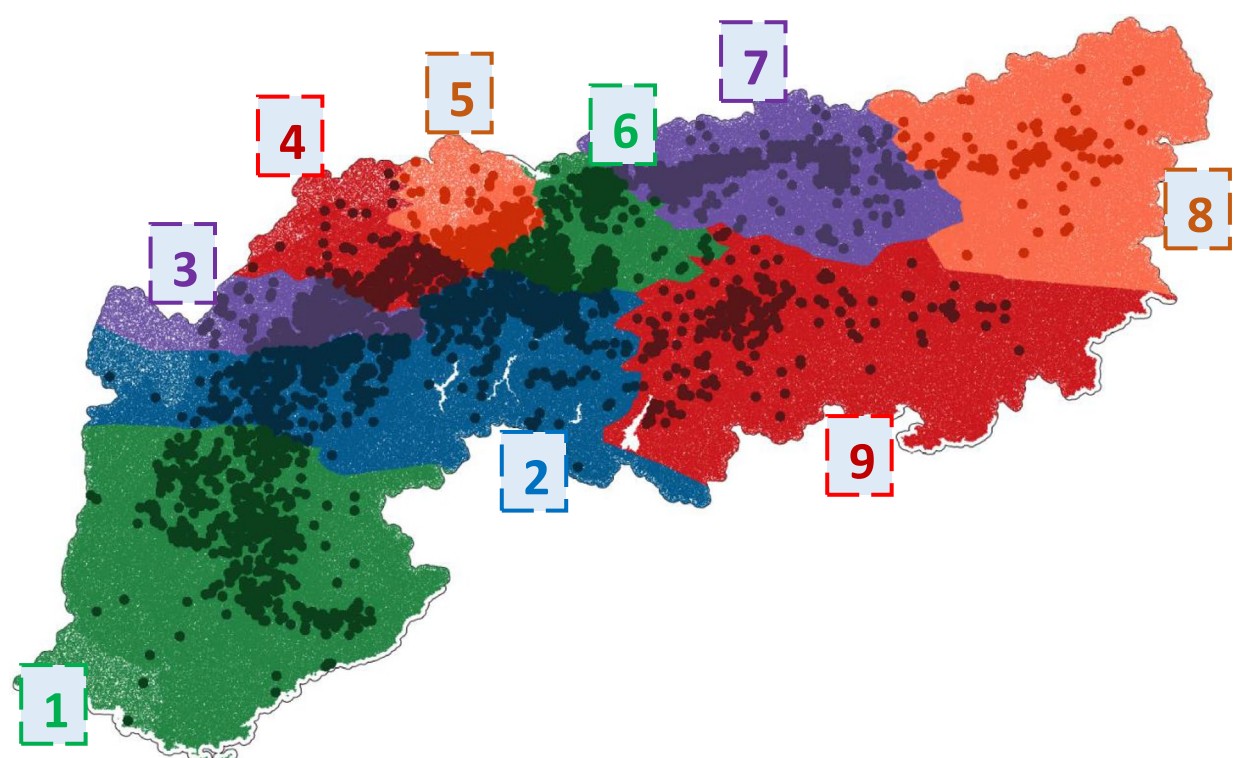

**Extended Data Fig. 7 | Example of spatial block split sampling for *Soldanella alpine*.** Following[103], its 8'755 observations (here highlighted in darker colours) were evenly partitioned into 9 blocks of 5 folds spatially stratified (-2 blocks per fold; numbers are indicated) using species observation coordinates, partitioning around medoids (PAM) clustering and the R package *cluster* (function *pam*)[185–187]. This allowed the number of observations to be spatially and numerically balanced within each independent fold. It is important to note that, even though the number of blocks was set to 10, this number may slightly vary per species conditional on the best balancing strategy of the PAM algorithm. Fold 1 (purple), 2 (red), 3 (orange), 4 (blue) and 5 (green) here contains 1'755, 1'538, 1'536, 1'976 and 1'950 species observations respectively. Sampled quadrature points and Flora Alpina presences/absences (here highlighted in lighter colours) were assigned to each independent block using k-nearest neighbour classification and the R package *class* (function *knn*)[188,189]. For every species and models (PPM), the same number of Flora Alpina presences/absences as quadrature points were sampled, to ensure balanced repartitioning among folds.

# Reporting Summary

## Statistics

For all statistical analyses, confirm that the following items are present in the figure legend, table legend, main text, or Methods section.

| n/a | Confirmed | |
|---|---|---|
| ☐ | ☒ | The exact sample size (*n*) for each experimental group/condition, given as a discrete number and unit of measurement |
| ☒ | ☐ | A statement on whether measurements were taken from distinct samples or whether the same sample was measured repeatedly |
| ☐ | ☒ | The statistical test(s) used AND whether they are one- or two-sided *Only common tests should be described solely by name; describe more complex techniques in the Methods section.* |
| ☐ | ☒ | A description of all covariates tested |
| ☐ | ☒ | A description of any assumptions or corrections, such as tests of normality and adjustment for multiple comparisons |
| ☐ | ☒ | A full description of the statistical parameters including central tendency (e.g. means) or other basic estimates (e.g. regression coefficient) AND variation (e.g. standard deviation) or associated estimates of uncertainty (e.g. confidence intervals) |
| ☐ | ☒ | For null hypothesis testing, the test statistic (e.g. *F*, *t*, *r*) with confidence intervals, effect sizes, degrees of freedom and *P* value noted *Give P values as exact values whenever suitable.* |
| ☒ | ☐ | For Bayesian analysis, information on the choice of priors and Markov chain Monte Carlo settings |
| ☒ | ☐ | For hierarchical and complex designs, identification of the appropriate level for tests and full reporting of outcomes |
| ☐ | ☒ | Estimates of effect sizes (e.g. Cohen's *d*, Pearson's *r*), indicating how they were calculated |

*Our web collection on statistics for biologists contains articles on many of the points above.*

## Software and code

Policy information about availability of computer code

| | |
|---|---|
| Data collection | R 4.1, python 3.9, GDAL/OGR 3.8.0, SAGA-GIS 7.2.0, Zonation 4 (CAZ/ABF) |
| Data analysis | R 4.1, python 3.9, GDAL/OGR 3.8.0, SAGA-GIS 7.2.0, Zonation 4 (CAZ/ABF), https://github.com/8Ginette8/wsl.biodiv, chelsa-cmip6 v1.0 python library, MigClim v1.6, V.Phylomaker v1.0, hillR v0.5.2, phyloregion v1.0.8, funrar v1.5.0, gbif.range v1.0, clValid v0.7, glmnet v4.1.8, MissMech v1.0.2, mice v3.16.0, cluster v2.1.4, and vegan v2.6.4 R packages. |

For manuscripts utilizing custom algorithms or software that are central to the research but not yet described in published literature, software must be made available to editors and reviewers. We strongly encourage code deposition in a community repository (e.g. GitHub). See the Nature Portfolio guidelines for submitting code & software for further information.

## Data

Policy information about availability of data

All manuscripts must include a data availability statement. This statement should provide the following information, where applicable:
- Accession codes, unique identifiers, or web links for publicly available datasets
- A description of any restrictions on data availability
- For clinical datasets or third party data, please ensure that the statement adheres to our policy

All data, code, and materials supporting the findings of this study are available in the EnviDat repository https://doi.org/10.16904/envidat.371, which provides options to download single (via WGET and singular FTP links) or every files (S3 access using the software Cyberduck or any other S3 clients) employed in this project.

# Human research participants

Policy information about studies involving human research participants and Sex and Gender in Research.

| | |
|---|---|
| Reporting on sex and gender | We did not use human participants for this study. |
| Population characteristics | We did not use human participants for this study. |
| Recruitment | We did not use human participants for this study. |
| Ethics oversight | We did not use human participants for this study. |

Note that full information on the approval of the study protocol must also be provided in the manuscript.

# Field-specific reporting

Please select the one below that is the best fit for your research. If you are not sure, read the appropriate sections before making your selection.

☐ Life sciences      ☐ Behavioural & social sciences      ☒ Ecological, evolutionary & environmental sciences

For a reference copy of the document with all sections, see nature.com/documents/nr-reporting-summary-flat.pdf

# Ecological, evolutionary & environmental sciences study design

All studies must disclose on these points even when the disclosure is negative.

| | |
|---|---|
| Study description | Here, we investigate the efficiency of the current European Alps' PAs network, and of its potential transnational expansion, in protecting the plant multifaceted diversity and uniqueness of this region at present and for the 2050 and 2080 horizons, under biologically informed (realistic) dispersal, two Shared Socioeconomic Pathways (SSPs), seven Global Circulation Model (GCMs) and two land cover (LC) change scenarios. Using an ensemble of species distribution models (SDMs) for 1,711 plant species at 100-m resolution, a high coverage database of species traits and two mega-phylogenies, we predicted future changes in species distributional range, multifaceted diversity and uniqueness in the study area. We then identified conservation priorities for all types of diversity, areas where these priorities are stable into the future (overlap in current and future priorities), and areas that are critical for expanding the current PA network to meet the 2030 targets while achieving a resilient and effective conservation network. |
| Research sample | (a) Sample (~6 millions) of plant observations (~4'000 species) compiled for the European Alps from many different sources including GBIF (https://doi.org/10.15468/dd.mb6jz) and 76 institutions. (b) Sample of ecological traits (4308 averaged traits) from 33 different institutions. (c) For the phylogeny, see these methods and sources: https://doi.org/10.1111/ecog.04434 |
| Sampling strategy | No sample-size calculation performed because we are here employing opportunistic data (from online sources and various institutions) already sampled. |
| Data collection | A large compilation of opportunistic/online/institution observations of plants, traits, phylogenies and community plots. |
| Timing and spatial scale | Plant species observed and/or sampled from 1980 to 2021 accurate to 11.1 meters. |
| Data exclusions | Observations accurate to > 11.1 meters were excluded. |
| Reproducibility | All attempts to reproduce the experiments were successful. |
| Randomization | Not relevant because modelling was used as a main method. |
| Blinding | Not relevant because a large amount of independent data were compiled. |

Did the study involve field work?      ☐ Yes      ☒ No

# Reporting for specific materials, systems and methods

We require information from authors about some types of materials, experimental systems and methods used in many studies. Here, indicate whether each material, system or method listed is relevant to your study. If you are not sure if a list item applies to your research, read the appropriate section before selecting a response.

## Materials & experimental systems

| n/a | Involved in the study |
|-----|----------------------|
| ☒ | Antibodies |
| ☒ | Eukaryotic cell lines |
| ☒ | Palaeontology and archaeology |
| ☒ | Animals and other organisms |
| ☒ | Clinical data |
| ☒ | Dual use research of concern |

## Methods

| n/a | Involved in the study |
|-----|----------------------|
| ☒ | ChIP-seq |
| ☒ | Flow cytometry |
| ☒ | MRI-based neuroimaging |

