## [Peer Review File · Nature Ecology & Evolution]

Peer Review Information

Journal: Nature Ecology & Evolution

Manuscript Title: Transnational conservation to anticipate future plant shifts in Europe

Corresponding author name(s): Yohann Chauvier

Editorial Notes:

Reviewer Comments & Decisions:

Decision Letter, initial version:

17th May 2023

Dear Dr Chauvier Mendes,

Your Article, "Transnational conservation to anticipate future plant shifts in Europe" has now been seen by 3 reviewers. You will see from their comments copied below that while they find your work of considerable potential interest, they have raised quite substantial concerns that must be addressed. In light of these comments, we cannot accept the manuscript for publication, but would be very interested in considering a revised version that addresses these concerns.

We therefore invite you to revise your manuscript taking into account all reviewer comments. Please highlight all changes in the manuscript text file. Please also ensure all data is fully available and accessible to the reviewers (see the point in Reviewer 1's comments).

* If you have not done so already please begin to revise your manuscript so that it conforms to our Article format instructions at <http://www.nature.com/natecolevol/info/final-submission>. Refer also to any guidelines provided in this letter.

2[REDACTED]

Nature Ecology & Evolution is committed to improving transparency in authorship. As part of our efforts in this direction, we are now requesting that all authors identified as 'corresponding author' on published papers create and link their Open Researcher and Contributor Identifier (ORCID) with their account on the Manuscript Tracking System (MTS), prior to acceptance. ORCID helps the scientific community achieve unambiguous attribution of all scholarly contributions. You can create and link your ORCID from the home page of the MTS by clicking on 'Modify my Springer Nature account'. For more information please visit <http://www.springernature.com/orcid>.

[REDACTED]

Reviewer expertise:

Reviewer #1: spatial conservation planning

Reviewer #2: species range shifts, conservation for climate change

Reviewer #3: vegetation distributions, land-use and climate change

Reviewer comments:

Reviewer #1 (Remarks to the Author):

Thank you very much for the opportunity to review manuscript NATECOLEVOL-23030689-T, "Transnational conservation to anticipate future plant shifts in Europe". This is a great topic to explore and I very much like the setup of this study. I have provided some detailed comments below. Overall I very much appreciate the amount of work that went into this study, but do have concerns that most

2readers will have a hard time following the presented approach as it is presented at this point. I think there needs to be quite a bit of tightening in the main text, but especially the methods section. I don't necessarily mean reducing the word count, although I do suggest considering moving some of the methodological details to supplementary materials. I am mostly recommending to keep the level of information consistent across sections. Currently, there are some sections that have very little information (e.g. Observations) whereas others (e.g. Environmental bias correction) provide details on R package function parameter settings.

I would further recommend making sure to use consistent terms throughout. One example that I provide details about below is on Species Distribution Models, Habitat Suitability maps, and Habitat Suitability map predictions. I would recommend picking one term, define it, and consistently use it throughout to make it easier for a reader to follow along.

I would highly recommend setting up the study in a way that makes it easy or easier for readers to follow what you do. A workflow schematic might help, some text on how the levels of complexity fit together would be ideal. This does not have to be in the main text, although it would help there. At least a supplementary figure would be useful. When going through the methods section I had a hard time keeping the components straight in my head and link details to the bigger picture at the right level. Making this easier for the reader would be a big gain in helping them understand this work I think.

Lastly, there was no discussion of any caveats of this work. I would highly recommend you add a discussion on caveats to the text. I find those sections crucially important to let readers know some of the potential shortcomings or issues with any work that's published.

Line 86+: I think some explanation of True Skill Statistic and Boyce Index are necessary if you call them out. I would not equate ~ 0.6 as 'very good' model performance.

Figure S1(a+b): there are at least 4 lines in each figure. The legend shows 3 lines and 2 lines. The way these figures are currently presented is not clear. I am not sure which colors correspond to scenarios and which to SSP's.

Figures S2: the figure legend does not sufficiently explain what a reader sees in the figures. What are the symbols on the left? What are all the figure legend acronyms mean? How should a reader interpret the values and ranges?

(I see that the symbols and acronyms are explained in legend of Figure 1. The value ranges are still not clear in that figure either)

Figure 3 is almost impossible to decipher. I have read the legend several times and stared at the figure, but can't say that I really understand what I am seeing here. I suggest rethinking this figure.

Local conservation vs regional conservation: In a spatial hierarchy local usually goes below regional, or a region covers more area than local. Consider rephrasing to make the level clearer.

Figure 4 is very busy. We have the study area plus elevation, then the scenario results across the study area and finally some scenario results across an elevation gradient and across countries. These are at least 3 figures in one and I would consider splitting them up and re-evaluating which figures should go in the main text.

Observation data (Table S2): The data sources need to be described. Acronyms alone are not sufficient. Are the data freely available? Are the cleaned data available to readers? At this point the reader does not have enough information to understand what data was used and how it was processed (other than GBIF data, for which processing is described in the methods section). Please describe data access and processing protocols for other data sources as well.

Land cover: were the data described used as is from the data provider or were any processing steps necessary. Please describe in text.

Bias covariate correction: was the observation density calculated on a species basis or across all observations? Please describe in text. If this was done across all species, is that a valid metric? Just because there are many observations of a common species in location A does not mean that observations of a rare species in location A would suffer an observation bias, does it? Please provide justification for the approach chosen.

Environmental bias correction: I appreciate the detailed description of the approach taken here. It would be good to have a consistent level of methodological detail throughout the methods section though. In this section R function argument details are presented and in other sections (e.g. observation data) most methods are omitted. I would ask the authors to maybe find a middle ground in the methods section and describe each section in enough detail that readers can make sense of them. Details such as model parameter settings could go into a supplemental material document or be linked to a well annotated script.

Species distribution models: please explain why point process models were used here and not more standard methods such as maximum entropy models (e.g. Maxent). It's important for readers to understand why this modelling technique was used here. As you describe point process models as belonging to the family of generalized linear models it would further be useful for readers to spend some time on the explanation of the need for 'absences' in the dataset. I am assuming these are the 'quadrature' points but am not entirely certain. As most readers will not be familiar with that term I recommend explaining it in more detail. One considerable challenge in modelling presence only data is the lack of 'absences' or 'non-detections'. I think it's important here to discuss this issue.

Further, please explain how the individual scenarios were generated, how many there were, and which variables were used for which model/scenario. At this point finding this information is not straight forward.

Evaluation: Here would be a good point to explain the two model performance parameters/test TSS and Boyce index in some detail. The reader needs to understand what a value of 0.3 means and this relates back to my earlier comment about 'very good' model performance at ~ 0.6 as well.

Projection:

I found the description in this section hard to follow. This is an important part of the methods section where the generation of map output for a range of scenarios is discussed. Could you think about a figure or schematic for supp mat that would sketch out the different models/maps/scenarios to make it easier for the reader to understand what's going on? It would also be helpful to use consistent terminology throughout would also be helpful. As an example, species distribution models, habitat suitability maps, habitats suitability probability maps are used as terms that could arguably describe the same thing. I would recommend picking one term and using it consistently throughout. My personal preference to be in line with a common use in the Maxent literature would be habitat suitability maps or models.

Line 428. "further processed (a) according to previous literature" is vague. What exactly did you do here?

Line 429. "with simple dispersal models" and mention of "realistic dispersal" in the next line is not clear. Were they simple dispersal models or realistic ones?

The sentence starting line 428 is not clear, what were the four additional suitability maps? 1711 species were kept is also not clear. Out of how many and why?

The term 'realistic dispersal' needs to be explained. What exactly do you mean by this? How is this realistic? Maybe using the term 'predicted dispersal' or 'restricted dispersal' would be more appropriate. Realistic is a strong statement and I am not sure you can really say the dispersal you predict is necessarily realistic.

Line 444. What was the best TSS threshold? You need to report that threshold somewhere (I would suggest supplementary material table).

Where MigClim parameters consistent across species or species specific?

System Conservation Planning – should be systematic conservation planning.

First question, why did you choose the Zonation software and not Marxan or prioritizr? Given that this is my main area of expertise I probably have more questions about this than the average reader.

Given that the choice of software matters considerably here, it would be good for the reader to understand your choice.

Full disclosure: I am one of the developers of the R package prioritizr.

Line 596 you mention 52 prioritization scenarios but Table S9 only hold 26. Why the difference?

The planning section is a critical component of this work and I would highly suggest to devote more than two paragraphs to this section. Is the approach chosen really the most appropriate for this study? Should conservation goals for individual species be considered (i.e. the main approach of using Marxan)? Why are planning scenarios investigated in isolation? Can you think about linking current and future plans? What are the consequences of running Zonation now and treating future Zonation runs independently? I see these as fundamental questions for this work. If you include dispersal in your species models, wouldn't it also make sense to consider protected area dispersal estimates as well? And if not quantitatively, at least discuss the consequences of not doing so?

5As for prioritization approaches, there are methods available that can incorporate phylogenetic diversity. See here: https://prioritizr.net/reference/add_max_phylo_div_objective.html
An example from the European Alps is Hanson et al. 2017.

Hanson, J. O., Rhodes, J. R., Riginos, C., & Fuller, R. A. (2017). Environmental and geographic variables are effective surrogates for genetic variation in conservation planning. *Proceedings of the National Academy of Sciences*, 114(48), 12755-12760.

Post-analysis questions are somewhat related to the systematic conservation planning questions. If the goal is to get to ~30% of the area, why not include this parameter right in the prioritization? Both Marxan and prioritizr have ways to include area constraints in the prioritization analysis.

The way you have setup your post-analysis section does make sense in general, but I do wonder why at least some of the post-analysis parts weren't accounted for in the conservation planning step already (see comments and questions above)?

Finally, I usually include a request to the authors to make their data and code as available as reasonable. I'm a big believer in the FAIR (Findable, Accessible, Interoperable and Reusable) principle for scientific data management and stewardship (e.g. <https://www.nature.com/articles/sdata201618>). Currently, I can see the EnviDat repository, but I am having a hard time accessing those data in a useful way and can't check whether I can replicate or reuse the work.

Best regards,
Richard Schuster, PhD

Director of Spatial Planning and Innovation, Nature Conservancy of Canada
Email: richard.schuster@natureconservancy.ca

Reviewer #2 (Remarks to the Author):

The paper "Transnational conservation to anticipate future plant shifts in Europe" incorporates current and projected species distribution models for Alps flora into spatial prioritizations to determine conservation priorities for flora by 2050 and 2080. The authors assess projected diversity changes in terms of taxonomic, phylogenetic, and functional diversity, endemism, and rarity (each). They find strong upslope shifts in Alps flora leading to increasing endemism and rarity and higher elevations. This has the effect of increasing threat for roughly 70% of the flora analyzed. By assessing priorities for protected area expansion alongside the current protected area network, they identify areas where local and regional (transnational) conservation is needed to optimally protect floral diversity across the

6Alps.

The authors have compiled an impressive dataset of species occurrences that have been extensively filtered and corrected to allow for robust species modeling. They also have obtained the best available datasets on climate and land cover projections, and high resolution maps of soil parameters. While I have a few questions pertaining to their methodological choices (see additional comments below), overall I felt the authors' approach was robust and sound.

Nevertheless, I found the main manuscript text lacked sufficient reference to the methods to provide the information necessary for interpreting and feeling confident about the results. I recognize that short-format papers, like in *Nature Ecology and Evolution*, require minimal in-text methods and that most methods be reserved until the end, but in general I felt that additional information on the input data, SDM approach, and prioritization in the text would have drastically improved the manuscript's clarity and readability.

But the bigger issue for me was that I did not find the results to substantially advance my understanding of expected species change patterns or conservation priorities in the region. There have been several older and more-recent studies assessing distributional shifts of plant species in the Alps, several using forecasted SDM and prioritization approaches, see, e.g.:

Parolo, G. and Rossi, G., 2008. Upward migration of vascular plants following a climate warming trend in the Alps. *Basic and Applied Ecology*, 9;

Casazza, G., Guerrina, M., Dagnino, D. and Minuto, L., 2023. Will natura 2000 european network of protected areas support conservation of Southwestern Alps endemic flora under future climate? *Biodiversity and Conservation*, 32;

Thom, D., Rammer, W., Laux, P., Smiatek, G., Kunstmann, H., Seibold, S. and Seidl, R., 2022. Will forest dynamics continue to accelerate throughout the 21st century in the Northern Alps? *Global Change Biology*, 28;

Dagnino, D., Guerrina, M., Minuto, L., Mariotti, M.G., Médail, F. and Casazza, G., 2020. Climate change and the future of endemic flora in the South Western Alps: relationships between niche properties and extinction risk. *Regional Environmental Change*, 20;

Schwager, P. and Berg, C., 2019. Global warming threatens conservation status of alpine EU habitat types in the European Eastern Alps. *Regional Environmental Change*, 19(8).

And there have been global prioritization studies that have focused on protecting biodiversity (including plants) alongside other important priorities (carbon and water) that have yielded important considerations for protected area expansion, see, e.g.:

Jung, M., Arnell, A., De Lamo, X., García-Rangel, S., Lewis, M., Mark, J., Merow, C., Miles, L., Ondo, I., Pironon, S. and Ravillious, C., 2021. Areas of global importance for conserving terrestrial biodiversity, carbon and water. *Nature Ecology & Evolution*, 5(11).

7What was missing for me in this paper was a clear expression of novelty as to how this paper advances our understanding, changes or expectations, or will guide conservation forward in the region (as that seems to be one of the intents).

The weakest aspect of this to me was the commentary on local and regional conservation and transnational conservation strategy. The discussion in these sections felt a bit generic and did not provide extensive guidance other than mentioning which countries should be responsible for protecting more land to reach 30% conservation targets. There was no discussion on how this could be achieved, or how the priorities from this univariate (i.e., plant biodiversity-focused) study might compare to other conservation priorities for the countries involved.

There was no conclusion to the manuscript, which left me wondering what the ultimate goal of the paper was. If it is to provide accurate and up-to-date information on species distribution change and composite variables of different aspects of functional, phylogenetic, and taxonomic diversity, then I think this takes a strong step forward, but the content matter might not be suitable for the journal. If it is to provide information on gaps in protected area networks and look forward to how gaps might change under climate change, then I think more care must be taken to put the results in a broader context and again feel that the focus of the study on plant diversity might lend itself better to a journal with a slightly narrow focus. But if the authors can provide a more compelling narrative as to the significance of the study and the importance of the results alongside much of the recent work on this important topic through further revisions, then I do think it could make a nice contribution in Nature Ecology and Evolution.

Additional comments:

Line 93 (and elsewhere) – Consider rewording “worst case SSP585 scenario” as SSP585 is not necessarily the worst case (e.g., SSP370 could be worse in a many ways). Perhaps use “more severe SSP585 scenario”?

Lines 118 and 121 – I think these sentences should read “the percentage of species losing/expanding the most of their suitable habitat...”

Figure 2 – y-axis should be “Elevation 95th Percentile (m)”. For the upper bar chart plot, I’m not sure why the y-axis is plotted as a secondary axis (on the right). Also, I don’t think it’s helpful to have the y-axis label is multicolored, and it’s a little confusing what the diagonal total % sp is referring to. And because the x-axis is scaled differently than that of the bottom contour plot, I think you should relabel the x-axis.

Methods:

Study area – Why are only IUCN category I-II PAs used in this analysis? It seems like that category III-VI PAs and PAs without categories provide some degree of biodiversity protection and should be considered here. Moreover, it seems reasonable that countries would include these PAs as contributing to meeting current protection targets, and thus excluding them in the analysis would mean that

8countries would be forced to meet higher targets and establish additional PAs where they otherwise might not have to. In fact, it's conceivable that the prioritization analysis could identify priority regions for expansion that are already protected in some capacity.

Environmental Data – Correlation – What was the common spatial resolution used for the projections given that the input data layers are in several different spatial resolutions? I assume 100 m given this is the resolution of the SDMs, but is that reasonable given that climate data are only available at 1 km?

Species distribution models – Calibration – There is research that shows that the choice of SDM algorithm can be quite influential in driving results (e.g., Muscatello, A., Elith, J. and Kujala, H., 2021. How decisions about fitting species distribution models affect conservation outcomes. *Conservation Biology*, 35; Hallgren, W., Santana, F., Low-Choy, S., Zhao, Y. and Mackey, B., 2019. Species distribution models can be highly sensitive to algorithm configuration. *Ecological Modelling*, 408; Thuiller, W., Guéguen, M., Renaud, J., Karger, D.N. and Zimmermann, N.E., 2019. Uncertainty in ensembles of global biodiversity scenarios. *Nature Communications*, 10), yet here the authors only use one SDM algorithm. The results would be much more convincing if multiple SDM algorithms were used, and also so that uncertainty to the choice of algorithm could also be assessed.

Reviewer #3 (Remarks to the Author):

The authors combined more than 6 million species-presence data with species distribution modeling approach to explore climate change-induced shifts in plant biodiversity and then assess the robustness of the conservation network in safeguarding the European Alps' flora by 2080. Their results suggested that warming could induce an overall shift in plant biodiversity, which highlighted the necessity for a transnational conservation strategy towards high-elevational plants. While this manuscript has the potential to make an important contribution to the biodiversity conservation over European alps, it requires extensive revisions to enhance the reliability of the main findings.

First, in terms of projecting climate change-induced shifts in plant biodiversity, one major challenge is to parameterize dispersal constraints in the species distribution modeling, and this is particularly true when working with a large number of species without accurate or realistic dispersal parameters. Therefore, the use of fixed or default parameters in species distribution modeling of "realistic" dispersal scenario could introduce great uncertainties in the rate and even direction of range shifts in the species distribution. The authors need to explore and discuss the potential uncertainties of dispersal parameters in the "realistic" scenario on the main results.

In addition, the authors considered dispersal scenarios for plants over European Alps but imposed a no-dispersal rule for all species at low elevations from surrounding regions. The omission of plant colonization from other regions would lead to an overestimation of the biodiversity loss, and undermine the validity of main findings. I would suggest to explore and discuss the inclusion of species dispersion from surrounding regions at low elevations in shaping the future plant biodiversity.

Second, the authors discussed the impact of land cover change on future plant biodiversity changes,

9but there is no actual data on this effect. Previous research has demonstrated an important impact of historical land cover changes on biodiversity changes (Garcés-Pastor et al., 2022). It is therefore important to compare the direction and rate of biodiversity changes due to land cover changes at least from the historical perspective with future climate change-induced biodiversity changes over the European Alps. Furthermore, there is clear evidence to show that the warming-induced upward shift of the life-form tree would compress the physical habitat area of alpine plants (Körner, 2012; Greenwood & Jump, 2018), which could potentially affect the species rarity as calculated in this study. The omission of tree upslope might induce an increase instead of a decline in endemism at the taxonomic level as shown in Figure 1.

Third, to achieve the biodiversity protection objectives of COP15, this study has increased the coverage of protected areas in the Alps to approximately 30%. However, it is still unclear how effective these protected areas are. By identifying species that are not currently covered by protected areas, targeted strategies can be developed to address these gaps and improve overall biodiversity conservation outcomes. Therefore, conducting a systematic assessment of gap species is crucial for effective biodiversity protection, especially for threatened and restricted range species (Rodrigues et al., 2004).

Lastly, it is expected that in a warmer climate, plants will migrate towards colder regions encompassing both high elevations and latitudes. A comprehensive understanding of future plant migration can be achieved by analyzing the two-dimensional changes in elevation and latitude together rather than separately. Conducting separate analyses across different sections may lead to fragmented knowledge about local conservation efforts. The authors suggest that optimizing conservation efforts in the Alps may require little European coordination; however, this contradicts the fact that plant migration is expected to occur across various regions and countries.

Minor comments:

1. Line 447: Please include sensitivity tests for various parameters utilized in simulating seed dispersal, such as `dispKernel`, barrier, and others.

2. In line 473, the authors describe a technique they developed to combine various seed dispersal scenarios. This technique incorporates 13 different scenarios that can be grouped into three categories: realistic, unrestricted, and no dispersal. Please add details and reasoning behind each step.

3. The results shown in Figure 1 demonstrate a significant variation in the endemism of alpine flora across functional, taxonomic, and phylogenetic dimensions. What are the underlying causes of this pattern?

References

Garcés-Pastor, S., Coissac, E., Lavergne, S., Schwörer, C., Theurillat, J. P., Heintzman, P. D., ... & Alsos, I. G. (2022). High resolution ancient sedimentary DNA shows that alpine plant diversity is associated with human land use and climate change. *Nature communications*, 13(1), 6559.

Greenwood, S., & Jump, A. S. (2014). Consequences of treeline shifts for the diversity and function of

10high altitude ecosystems. *Arctic, Antarctic, and Alpine Research*, 46(4), 829-840.

Körner, C. (2012). *Alpine treelines: functional ecology of the global high elevation tree limits*. Springer Science & Business Media.

Rodrigues, A. S., Andelman, S. J., Bakarr, M. I., Boitani, L., Brooks, T. M., Cowling, R. M., ... & Yan, X. (2004). Effectiveness of the global protected area network in representing species diversity. *Nature*, 428(6983), 640-643.

*****END*****

Author Rebuttal to Initial comments

Reviewer #1 (Remarks to the Author):

1-1) Thank you very much for the opportunity to review manuscript NATECOLEVOL-23030689-T, “Transnational conservation to anticipate future plant shifts in Europe”. This is a great topic to explore and I very much like the setup of this study. I have provided some detailed comments below. Overall I very much appreciate the amount of work that went into this study, but do have concerns that most readers will have a hard time following the presented approach as it is presented at this point. I think there needs to be quite a bit of tightening in the main text, but especially the methods section. I don’t necessarily mean reducing the word count, although I do suggest considering moving some of the methodological details to supplementary materials. I am mostly recommending to keep the level of information consistent across sections. Currently, there are some sections that have very little information (e.g. Observations) whereas others (e.g. Environmental bias correction) provide details on R package function parameter settings.

Thanks for these helpful comments. We have reworked the manuscript and tried to better balance detail and clarity across sections. In particular, we have reworked the methods and written the essential methods in the main test and transferred too detailed parts (or technical) to the appendix. We think these modifications improve and address the overall complexity that you have pointed out through some of your comments.

111-2) I would further recommend making sure to use consistent terms throughout. One example that I provide details about below is on Species Distribution Models, Habitat Suitability maps, and Habitat Suitability map predictions. I would recommend picking one term, define it, and consistently use it throughout to make it easier for a reader to follow along.

Thanks, we now see the inconsistency and have corrected it throughout the manuscript. Since species distribution (model) is a term usually more used in Europe, we have decided to go for this one. We also did a sanity check for other terms.

1-3) I would highly recommend setting up the study in a way that makes it easy or easier for readers to follow what you do. A workflow schematic might help, some text on how the levels of complexity fit together would be ideal. This does not have to be in the main text, although it would help there. At least a supplementary figure would be useful. When going through the methods section I had a hard time keeping the components straight in my head and link details to the bigger picture at the right level. Making this easier for the reader would be a big gain in helping them understand this work I think.

We understand, and a schematic workflow is indeed a good idea to keep a level of method transparency and clarity for this work. We have therefore added a Fig. S1 summarizing the workflow of this paper. Thank you.

1-4) Lastly, there was no discussion of any caveats of this work. I would highly recommend you add a discussion on caveats to the text. I find those sections crucially important to let readers know some of the potential shortcomings or issues with any work that's published.

Caveats are now discussed in the new last section of the manuscript (L282-329). We here address the challenges to achieve comprehensive transnational conservation, such as the need to study a bigger extent than our study region to consider future migration from outside the alps, the need to also include within conservation planning ecosystem services (regulatory and provision) and species interaction richness (Pollock et al. 2020, O'Connor et al. 2021, Gaüzère et al. 2022).

1-5) Line 86+: I think some explanation of True Skill Statistic and Boyce Index are necessary if you call them out. I would not equate ~ 0.6 as 'very good' model performance.

You are completely right, we actually forgot to briefly explain it. Information was added in methods (L460-464). Thank you.

1-6) Figure S1(a+b): there are at least 4 lines in each figure. The legend shows 3 lines and 2 lines. The way these figures are currently presented is not clear. I am not sure which colors correspond to scenarios and which to SSP's.

This figure is now figure (S2ad). There are in fact five curves (current, 2050-SSP245, 2050-SSP585, 2080-SSP245 and 2080-SSP585). Following your recommendations, we changed the colour contrasts to better decipher the different expansions. The visual has improved for some of the sub-plots, although due to some overlaps, all curves are not all sometimes visible. We have added a sentence in the legend to specify it and be more transparent for the reader. Also, we have added results (L96) on no dispersal to show that if species would not be able to move, losses of multifaceted diversity would be higher.

1-7) Figures S2: the figure legend does not sufficiently explain what a reader sees in the figures. What are the symbols on the left? What are all the figure legend acronyms mean? How should a reader interpret the values and ranges?

(I see that the symbols and acronyms are explained in legend of Figure 1. The value ranges are still not clear in that figure either)

This figure is now figure (S4a). Yes, thank you, we have addressed these changes.

1-8) Figure 3 is almost impossible to decipher. I have read the legend several times and stared at the figure, but can't say that I really understand what I am seeing here. I suggest rethinking this figure.

This figure shows conservation hotspots from the systematic conservation algorithm, considering current, 2050 and 2080. Blue stands for important conservation areas for present, whereas green and

red stands for the same information for 2050 and 2080 respectively. The other colours that are found represent conservation overlaps between timelines. We have added contours to delineate areas already protected (as in Fig. 4) and improved the figure legend, along with the main text (explaining local vs regional conservation strategy) for better clarity (L165-174 & L205-210).

1-9) Local conservation vs regional conservation: In a spatial hierarchy local usually goes below regional, or a region covers more area than local. Consider rephrasing to make the level clearer.

Yes, we realize that we have misworded some part of the text. We have corrected this issue and made the distinction between local and regional clearer (same lines as above).

1-10) Figure 4 is very busy. We have the study area plus elevation, then the scenario results across the study area and finally some scenario results across an elevation gradient and across countries. These are at least 3 figures in one and I would consider splitting them up and re-evaluating which figures should go in the main text.

We see the point. Yet, we believe that all elements are quite important for the reader to understand what is spatially (A) and administratively (B) happening in terms of reserve network expansion (top 20%) when maximising the diversity for the whole region (gamma diversity). We could remove panel (A left), but non-European readers would surely need a map of how the countries are distributed to orient their eyes relative to the described changes in (B).

1-11) Observation data (Table S2): The data sources need to be described. Acronyms alone are not sufficient. Are the data freely available? Are the cleaned data available to readers? At this point the reader does not have enough information to understand what data was used and how it was processed (other than GBIF data, for which processing is described in the methods section). Please describe data access and processing protocols for other data sources as well.

We understand your concern and realise that this section was not clear enough. We have added information quickly describing the acronyms and restructured the whole *Observations* section by making

a clear distinction between the sensitive dataset (only accessible upon reasonable requests due to data policies we don't have any control on) and the GBIF dataset (accessible on the Envidat depository).

1-12) Land cover: were the data described used as is from the data provider or were any processing steps necessary. Please describe in text.

Indeed, while the original ECOCHANGE outputs included the whole Alps, the HERCULES outputs did not cover Switzerland. Prof. Dr. Peter Verburg has been part of the HERCULES project for a while now, therefore, using the same code and methodology, we did re-run new LC change models for the whole European Alps with his help. We have clarified it in the methods (L394-398).

1-13) Bias covariate correction: was the observation density calculated on a species basis or across all observations? Please describe in text. If this was done across all species, is that a valid metric? Just because there are many observation of a common species in location A does not mean that observations of a rare species in location A would suffer an observation bias, does it? Please provide justification for the approach chosen.

The observation density was calculated across all observations and employed in the models following (Chauvier et al. 2021). The justification behind is based on assuming and summarizing a general pattern of observer bias across the dataset that would potentially summarize most species observation patterns. Of course, all species (e.g., rare species) don't necessarily have biased observations, or if they do, their patterns might not be explained by this observation density (or by the density of roads and cities), in which case the model would simply not fit the observations against the bias covariates, hence applying no corrections. Interestingly, applying instead such a correction on a species basis might sound tempting, however, most often, the model response would just be cancelled out since it would fit species observations against their own density (unless an external observational dataset is available). We have therefore tried to make the text clearer by summarising this information and making a clear reference to it (L407-422).

1-14) Environmental bias correction: I appreciate the detailed description of the approach taken here. It would be good to have a consistent level of methodological detail throughout the methods section

15though. In this section R function argument details are presented and in other sections (e.g. observation data) most methods are omitted. I would ask the authors to maybe find a middle ground in the methods section and describe each section in enough detail that readers can make sense of them. Details such as model parameter settings could go into a supplemental material document or be linked to a well annotated script.

Yes, we knew we would need to change the format of the methods to make them more concise and clearer for the reader. We have now improved the method section by homogenizing and shortening it in the main text, while referring to the supplementary materials if the reader wants more technical information. We did it for the *observations, soil, Environmental Bias Correction, Calibration, Diversity, Uniqueness* and *Post-analyses* sections.

1-15) Species distribution models: please explain why point process models were used here and not more standard methods such as maximum entropy models (e.g. Maxent). Its important for readers to understand why this modelling technique was used here. As you describe point process models as belonging to the family of generalized linear models it would further be useful for readers to spend some time on the explanation of the need for ‘absences’ in the dataset. I am assuming these are the ‘quadrature’ points but am not entirely certain. As most readers will not be familiar with that term I recommend explaining it in more detail. One considerable challenge in modelling presence only data is the lack of ‘absences’ or ‘non-detections’. I think it’s important here to discuss this issue.

You are right and reviewer 2 has made a similar comment. Interestingly, Maxent is a point-process model (Renner and Warton 2013) but the machinery is hidden in the software without much flexibility and detail. We thus used a very similar PPM but implemented in another package that automatically choose the number and location of ‘background points’ or ‘quadrature points’ (to avoid unstable model). This is a preferable solution than Maxent (Renner et al. 2015). We have therefore fully explained the rationale behind choosing PPMs for our models (L445-452), i.e., the presence of a natural framework to automatically choose background points, the clarity of the model response, an intuitive way to correct observer bias, no truncated species response curve due to a potential lack of background points and easy implementation of lasso.

1-16) Further, please explain how the individual scenarios were generated, how many there were, and which variables were used for which model/scenario. At this point finding this information is not straight forward.

We have now shortened the methods and change the whole *Projection* section accordingly. Also, additional information was added in the new workflow of Fig. S1.

1-17) Evaluation: Here would be a good point to explain the two model performance parameters/test TSS and Boyce index in some detail. The reader needs to understand what a value of 0.3 means and this relates back to my earlier comment about 'very good' model performance at ~0.6 as well.

You have made the according changes in combination with comment (1-5).

1-18) Projection:

I found the description in this section hard to follow. This is an important part of the methods section where the generation of map output for a range of scenarios is discussed. Could you think about a figure or schematic for supp mat that would sketch out the different models/maps/scenarios to make it easier for the reader to understand what's going on? It would also be helpful to use consistent terminology throughout would also be helpful. As an example, species distribution models, habitat suitability maps, habitats suitability probability maps are used as terms that could arguably describe the same thing. I would recommend picking one term and using it consistently throughout. My personal preference to be in line with a common use in the Maxent literature would be habitat suitability maps or models.

The whole *Projection* section has been entirely improved and is now clearer. Also, we adapted the schematic (Fig. S1) in the appendix accordingly.

1-19) Line 428. "further processed (a) according to previous literature" is vague. What exactly did you do here? Line 429. "with simple dispersal models" and mention of "realistic dispersal" in the next line is not clear. Were they simple dispersal models or realistic ones? The sentence starting line 428 is not clear, what were the four additional suitability maps? 1711 species were kept is also not clear. Out of how many and why?

17We indeed realized that the *Projection* section was unclear. As said earlier, this part has been completely changed. We hope that it helps understanding our methods.

1-20) The term ‘realistic dispersal’ needs to be explained. What exactly do you mean by this? How is this realistic? Maybe using the term ‘predicted dispersal’ or ‘restricted dispersal’ would be more appropriate. Realistic is a strong statement and I am not sure you can really say the dispersal you predict is necessarily realistic.

We have changed the wording to ‘limited dispersal’ since we are also talking about ‘unlimited dispersal’.

1-21) Line 444. What was the best TSS threshold? You need to report that threshold somewhere (I would suggest supplementary material table).

We have clarified this part (L474-476) and explained that, in order to convert our SDM intensity maps to binary maps, we used for each species the mean of the thresholds that maximise the TSS (‘maximum TSS’) of all retained calibrated models.

1-22) Where MigClim parameters consistent across species or species specific?

Important ecological parameters (such as barriers or dispersal in meters) are species specific and based on species ecological information previously compiled in the “alpine” literature. Other parameters cannot be retrieved by species because unknown and relatively stochastic, so we have set those according to previous literatures that have used the same dispersal models for plants (e.g., Long distance dispersal, propagule production). We have added in the appendix a new table (S7) describing all that for more clarity and linked to the *Projection* section. Also, we discuss in the last section of the main manuscript how changing some parameters (plant maturity age) could influence the rate of plant migration.

1-23) System Conservation Planning – should be systematic conservation planning.

First question, why did you choose the Zonation software and not Marxan or prioritizr? Given that this is my main area of expertise I probably have more questions about this than the average reader. Given that the choice of software matters considerably here, it would be good for the reader to understand your choice.

Full disclosure: I am one of the developers of the R package prioritizr.

We chose to use Zonation over Marxan or prioritizr for the following reasons: (1) we as a co-author group are very familiar with Zonation including the algorithm and behaviour in a range of scenarios, especially when applied to many species. We are also collaborators and in close contact with the developers of Zonation so we can easily troubleshoot any issue. (2) Following from this, we have previously used Zonation to maximize 'local' versus 'regional' diversity (Pollock et al. 2017) and we applied this same framework in this paper. As these solutions can produce very different results, we wanted to ensure we captured that range of solutions in this paper. (3) We have not agreed upon species-based targets across all species and therefore sought to minimize the issues with thresholding in the sense that we would not be confident that our targets would represent 'adequately protected'. While we are aware there is a Zonation-like solution in prioritizr, we are not as familiar with it. (4) Zonation has been used in the past to use distributions that are projected for the present and the future (e.g. Kujala et al. 2013). Last, Zonation, because it is more of a heuristic than full optimization, processes large fine-scale rasters very quickly. This is essential given the number of runs and species involved in this study. We have added and further argued our choices in the *Zonation* section (L527-537). Thank you.

1-24) Line 596 you mention 52 prioritization scenarios but Table S9 only hold 26. Why the difference?

We have modified the legend in Table (now S10), to more clearly specify that we have indeed in total 52 prioritization scenarios (or 26 per prioritization algorithm; CAZ and ABF).

1-25) The planning section is a critical component of this work and I would highly suggest to devote more than two paragraphs to this section. Is the approach chosen really the most appropriate for this study? Should conservation goals for individual species be considered (i.e. the main approach of using Marxan)? Why are planning scenarios investigated in isolation? Can you think about linking current and future plans? What are the consequences of running Zonation now and treating future Zonation runs

19independently? I see these as fundamental questions for this work. If you include dispersal in your species models, wouldn't it also make sense to consider protected area dispersal estimates as well? And if not quantitatively, at least discuss the consequences of not doing so?

As for prioritization approaches, there are methods available that can incorporate phylogenetic diversity. See here: https://prioritizr.net/reference/add_max_phylo_div_objective.html
An example from the European Alps is Hanson et al. 2017.

Hanson, J. O., Rhodes, J. R., Riginos, C., & Fuller, R. A. (2017). Environmental and geographic variables are effective surrogates for genetic variation in conservation planning. *Proceedings of the National Academy of Sciences*, 114(48), 12755-12760.

Following comment **(1-23)** we have improved the *Zonation* section in methods. Rationale behind why *Zonation* was chosen were added, and justification of our isolated SCP approach per scenarios explained (L527-537 & 541-550). For the latter, one main reason for this approach was to be able to disentangle and assess conservation priorities between current and future scenarios. Having an overall SCP scenario for all timelines (including all biodiversity features regardless) would not have allowed to assess the amount of spatial and administrative conservation between current and future conditions and its evolution across time and along elevation. Moreover, future species distributions are only projections based on expected climate, dispersal, and land cover scenarios. A lot of uncertainties drive these expectations and analysing them separately surely provide more transparency for us and the reader.

1-25) Post-analysis questions are somewhat related to the systematic conservation planning questions. If the goal is to get to ~30% of the area, why not include this parameter right in the prioritization? Both Marxan and prioritizr have ways to include area constraints in the prioritization analysis.

Yes, it is related to comment **(1-23)**. We still prefer to have the entire landscape ranked as it can be useful beyond the 30% cutoff in post-hoc analyses. The range of values allows us to be flexible and distinguish between the 5% most important, 10% most important etc. This was a very large analysis with the many detailed SDMs so we decided against comparing target-based conservation planning with *Zonation* as this comparison has been made in the past. We have now added a bit more explanation related to that in the SCP methods (L527-537). *Zonation* is a standard software used the most in applied conservation in Europe so there are also advantageous is recognition and communication to stakeholders beyond this academic paper.

1-26) The way you have setup your *Post-analysis* section does make sense in general, but I do wonder why at least some of the post-analysis parts weren't accounted for in the conservation planning step already (see comments and questions above)?

Yes, it is a very good question, and we should have justified it. To be fairly honest, we had in the first draft considered the phylogenetic and functional conservation aspect by running SCPs on the binary distribution of the phylogenetic and functional branches (from our dendrograms) of the European Alps. After some considerations, we have updated this part to a more straightforward and intuitive way of including multifaceted diversity into conservation planning (phylogenetic and functional uniqueness as weights). However, as you mentioned, we kept this binary distribution of branches to use it in the post-analyses and calculate the accumulation curves from Fig. S2ad that gave us very useful information on the amount of multifaceted diversity loss that could be expected by 2080. Choices were justified the *Zonation* section (L541-550).

1-27) Finally, I usually include a request to the authors to make their data and code as available as reasonable. I'm a big believer in the FAIR (Findable, Accessible, Interoperable and Reusable) principle for scientific data management and stewardship (e.g. <https://www.nature.com/articles/sdata201618>). Currently, I can see the EnviDat repository, but I am having a hard time accessing those data in a useful way and can't check whether I can replicate or reuse the work.

As specified on the right panel of the repository, Envidat proposes two ways of downloading data. Either singular files by ticking desired checkboxes - usually the case when the user only wants specific files such as some data - or all the data available from the repository (or in specific folders) by downloading a S3 access file and opening it with Cyberduck (free software). We have also tried to access the whole data via Cyberduck, and we realized that we could not get to it either. The problem was actually on Envidat's end. Due to a recent Cyberduck updates, they had to make some changes. It is now working. Do not hesitate to notify us again if the problem persists...

Best regards,
Richard Schuster, PhD

Director of Spatial Planning and Innovation, Nature Conservancy of Canada
Email: richard.schuster@natureconservancy.ca

Reviewer #2 (Remarks to the Author):

2-1) The paper “Transnational conservation to anticipate future plant shifts in Europe” incorporates current and projected species distribution models for Alps flora into spatial prioritizations to determine conservation priorities for flora by 2050 and 2080. The authors assess projected diversity changes in terms of taxonomic, phylogenetic, and functional diversity, endemism, and rarity (each). They find strong upslope shifts in Alps flora leading to increasing endemism and rarity and higher elevations. This has the effect of increasing threat for roughly 70% of the flora analyzed. By assessing priorities for protected area expansion alongside the current protected area network, they identify areas where local and regional (transnational) conservation is needed to optimally protect floral diversity across the Alps. The authors have compiled an impressive dataset of species occurrences that have been extensively filtered and corrected to allow for robust species modeling. They also have obtained the best available datasets on climate and land cover projections, and high resolution maps of soil parameters. While I have a few questions pertaining to their methodological choices (see additional comments below), overall I felt the authors’ approach was robust and sound. Nevertheless, I found the main manuscript text lacked sufficient reference to the methods to provide the information necessary for interpreting and feeling confident about the results. I recognize that short-format papers, like in Nature Ecology and Evolution, require minimal in-text methods and that most methods be reserved until the end, but in general I felt that additional information on the input data, SDM approach, and prioritization in the text would have drastically improved the manuscript’s clarity and readability.

Yes, thank you for your comment, we realise that now. We have now summarized in the beginning of each section (in the main text) which methods and data inputs were used to obtain the described results (L90-94, L129-131, L169-171, L208-210). We hope that this increased the readability of the manuscript.

2-2) But the bigger issue for me was that I did not find the results to substantially advance my understanding of expected species change patterns or conservation priorities in the region. There have been several older and more-recent studies assessing distributional shifts of plant species in the Alps, several using forecasted SDM and prioritization approaches, see, e.g.:

Parolo, G. and Rossi, G., 2008. Upward migration of vascular plants following a climate warming trend in

22the Alps. *Basic and Applied Ecology*, 9;

Casazza, G., Guerrina, M., Dagnino, D. and Minuto, L., 2023. Will natura 2000 european network of protected areas support conservation of Southwestern Alps endemic flora under future climate? *Biodiversity and Conservation*, 32;

Thom, D., Rammer, W., Laux, P., Smiatek, G., Kunstmann, H., Seibold, S. and Seidl, R., 2022. Will forest dynamics continue to accelerate throughout the 21st century in the Northern Alps? *Global Change Biology*, 28;

Dagnino, D., Guerrina, M., Minuto, L., Mariotti, M.G., Médail, F. and Casazza, G., 2020. Climate change and the future of endemic flora in the South Western Alps: relationships between niche properties and extinction risk. *Regional Environmental Change*, 20;

Schwager, P. and Berg, C., 2019. Global warming threatens conservation status of alpine EU habitat types in the European Eastern Alps. *Regional Environmental Change*, 19(8).

And there have been global prioritization studies that have focused on protecting biodiversity (including plants) alongside other important priorities (carbon and water) that have yielded important considerations for protected area expansion, see, e.g:

Jung, M., Arnell, A., De Lamo, X., García-Rangel, S., Lewis, M., Mark, J., Merow, C., Miles, L., Ondo, I., Pironon, S. and Ravilious, C., 2021. Areas of global importance for conserving terrestrial biodiversity, carbon and water. *Nature Ecology & Evolution*, 5(11).

What was missing for me in this paper was a clear expression of novelty as to how this paper advances our understanding, changes or expectations, or will guide conservation forward in the region (as that seems to be one of the intents).

The weakest aspect of this to me was the commentary on local and regional conservation and transnational conservation strategy. The discussion in these sections felt a bit generic and did not provide extensive guidance other than mentioning which countries should be responsible for protecting more land to reach 30% conservation targets. There was no discussion on how this could be achieved, or how the priorities from this univariate (i.e., plant biodiversity-focused) study might compare to other conservation priorities for the countries involved.

We have improved this part by further homogenising the manuscript and putting the results in the broader context that is showing that the current EU network protect less multifaceted diversity than our SCP recommendations, promoting the EU next transnational framework, what it needs to protect (multifaceted diversity and their limited migration), and the further perspectives and challenges to achieve this (e.g. large observational & dispersal dataset often lacking for other taxa than plants). Two new sections (EU conservation perspectives & Challenges) were added at the end of the manuscript for this purpose. Thank you for pointing this out, the manuscript has improved now.

2-3) There was no conclusion to the manuscript, which left me wondering what the ultimate goal of the paper was. If it is to provide accurate and up-to-date information on species distribution change and composite variables of different aspects of functional, phylogenetic, and taxonomic diversity, then I think this takes a strong step forward, but the content matter might not be suitable for the journal. If it is to provide information on gaps in protected area networks and look forward to how gaps might change under climate change, then I think more care must be taken to put the results in a broader context and again feel that the focus of the study on plant diversity might lend itself better to a journal with a slightly narrow focus. But if the authors can provide a more compelling narrative as to the significance of the study and the importance of the results alongside much of the recent work on this important topic through further revisions, then I do think it could make a nice contribution in Nature Ecology and Evolution.

As specified in our previous comment, the narrative has now been improved, as well as the general message of the manuscript. Thank you for that.

Additional comments:

2-4) Line 93 (and elsewhere) – Consider rewording “worst case SSP585 scenario” as SSP585 is not necessarily the worst case (e.g., SSP370 could be worse in a many ways). Perhaps use “more severe SSP585 scenario”?

We have made the according changes (L100 & 138).

2-5) Lines 118 and 121 – I think these sentences should read “the percentage of species losing/expanding the most of their suitable habitat...”

We have made the according changes (L137, 133-134). Thank you.

2-6) Figure 2 – y-axis should be “Elevation 95th Percentile (m)”. For the upper bar chart plot, I’m not sure why the y-axis is plotted as a secondary axis (on the right). Also, I don’t think it’s helpful to have the y-axis label is multicolored, and it’s a little confusing what the diagonal total % sp is referring to. And because the x-axis is scaled differently than that of the bottom contour plot, I think you should relabel the x-axis.

We have made some changes without removing the essence of the plot and modified the legend for more clarity.

Methods:

2-7) Study area – Why are only IUCN category I-II PAs used in this analysis? It seems like that category III-VI PAs and PAs without categories provide some degree of biodiversity protection and should be considered here. Moreover, it seems reasonable that countries would include these PAs as contributing to meeting current protection targets, and thus excluding them in the analysis would mean that countries would be forced to meet higher targets and establish additional PAs where they otherwise might not have to. In fact, it’s conceivable that the prioritization analysis could identify priority regions for expansion that are already protected in some capacity.

Yes, it is indeed a very interesting question. To be fairly honest, we ran our first analyses with IUCN I-VI (including equivalent PAs) following the same reasoning. However, after a lot of discussions, we concluded that it would be more relevant to simulate the new reserve network based on IUCN I-II categories. Overall, we know that the current reserve network in Europe (including the European Alps) is outdated compared to what we currently know about what composed diversity and its contribution to ecosystem functions (Kukkala et al. 2016, Hoffmann et al. 2018). And this actually also applies to the Natura 2000 network - largest coordinated network worldwide (Orlikowska et al. 2016) - which was only originally structured to protect habitat (from the Habitat Directive), rare species and migratory birds (so no multifaceted diversity or Beta complementarity between sites). Knowing that IUCN III-VI do not strictly protect biodiversity due to their inherent characteristic of “managed” zones, they are known to harbour and protect less biodiversity than the other categories. On the contrary, IUCN I-II PAs are

25considered as cornerstones of the EU network. Due to their (still) significant benefits to biodiversity conservation and large sizes (Watson et al. 2014, Gray et al. 2016, Hoffmann et al. 2018), they might be efficient in implementing future conservation insights (Hoffmann et al. 2018). For all these reasons, we considered the IUCN I-II + Natura 2000 network as to be a fair and realistic backbone (~18% of the European Alps) to predict an updated and adaptative novel reserve network, and to demonstrate that extending this network to ~30% of the landscape protects current and future multifaceted diversity way more efficiently than the IUCN I-VI network (+ equivalent PAs), which also covers ~30% of the landscape. This also relates to one of the comments from reviewer 3 (3-4), and we have therefore added this information in two new sections where transnational conservation perspectives and challenges are discussed. Thank you for pointing this out.

2-8) Environmental Data – Correlation – What was the common spatial resolution used for the projections given that the input data layers are in several different spatial resolutions? I assume 100 m given this is the resolution of the SDMs, but is that reasonable given that climate data are only available at 1 km?

Projections were first done at a 100x100m resolution, and intensity values (or abundances) then aggregated to 1x1-km as this has been shown to give better representations of diversity distribution (Thuiller et al. 2015, Chauvier et al. 2022). The fact of projecting everything to 100x100-m is not a problem as long as the original resolution of the environments represent well the conditions we are analysing (categorical land cover and meso-climate at both 1x1-km). Therefore, the practice of multiple-grain SDMs is quite common and admitted across the literature (Bellamy et al. 2013, Connor et al. 2018, Mertes and Jetz 2018).

2-9) Species distribution models – Calibration – There is research that shows that the choice of SDM algorithm can be quite influential in driving results (e.g., Muscatello, A., Elith, J. and Kujala, H., 2021. How decisions about fitting species distribution models affect conservation outcomes. *Conservation Biology*, 35; Hallgren, W., Santana, F., Low-Choy, S., Zhao, Y. and Mackey, B., 2019. Species distribution models can be highly sensitive to algorithm configuration. *Ecological Modelling*, 408; Thuiller, W., Guéguen, M., Renaud, J., Karger, D.N. and Zimmermann, N.E., 2019. Uncertainty in ensembles of global biodiversity scenarios. *Nature Communications*, 10), yet here the authors only use one SDM algorithm. The results would be much more convincing if multiple SDM algorithms were used, and also so that uncertainty to the choice of algorithm could also be assessed.

It's indeed a good question. PPM was chosen because of its simplicity and intuitively which gives larger benefits compared to more classical SDM approaches (Warton and Shepherd 2010, Renner et al. 2015). This algorithm is not yet as much used compared to ensemble approach, but an increasing number of studies implements the individual algorithm in their research (Merow et al. 2017, El-Gabbas and Dormann 2018, Thuiller et al. 2018, Renner et al. 2019, Bonnet-Lebrun et al. 2020, Botella et al. 2020, Carlson et al. 2022). PPM is not entirely comparable to other more classical SDM approaches, as it possesses a more natural framework to automatically choose the number and location of background points (to avoid unstable model), associated with a lasso implementation. Therefore, combining PPM with other “unnatural” SDM methods in an ensemble approach may methodologically be contested. We have therefore fully explained the rationale behind choosing PPMs for our models (L438-452), i.e., the presence of a natural framework to automatically choose background points, the clarity of the model response, an intuitive way to correct observer bias, no truncated response curve due to a potential lack of background points and an easy implementation of lasso.

Reviewer #3 (Remarks to the Author):

3-1) The authors combined more than 6 million species-presence data with species distribution modeling approach to explore climate change-induced shifts in plant biodiversity and then assess the robustness of the conservation network in safeguarding the European Alps' flora by 2080. Their results suggested that warming could induce an overall shift in plant biodiversity, which highlighted the necessity for a transnational conservation strategy towards high-elevational plants. While this manuscript has the potential to make an important contribution to the biodiversity conservation over European alps, it requires extensive revisions to enhance the reliability of the main findings.

First, in terms of projecting climate change-induced shifts in plant biodiversity, one major challenge is to parameterize dispersal constraints in the species distribution modeling, and this is particularly true when working with a large number of species without accurate or realistic dispersal parameters.

Therefore, the use of fixed or default parameters in species distribution modeling of “realistic” dispersal scenario could introduce great uncertainties in the rate and even direction of range shifts in the species distribution. The authors need to explore and discuss the potential uncertainties of dispersal parameters in the “realistic” scenario on the main results.

It is indeed a point we had not raised in the previous version of the manuscript. We have corrected this and added some elements of discussion in the last *Challenges* section. Overall, we however think that we were not so clear on how the dispersal parameters were used. The most important dispersal parameters are *dispKernel* and *barriers*. They are both species specific and based on known ecological information only available for the European Alps Flora (see new *Projection* section and table S7). Other parameters (e.g., LDD, dispersion curve) were indeed set the same for all species. However, this value was carefully chosen according to the known literature (of the creators of the MigClim package), since this cannot be done for each species because the information does not simply exist and is very hard to predict. This is also partially the case for 'iniMatAge'. This parameter determines after how many years the species may disperse, therefore partly driving the species rate of dispersal. Although the information per species could not be retrieved, it is true that setting a common values of age maturity to all taxonomic groups (e.g., herbs, trees, shrubs...) is not realistic. We have tried to choose a compromise of 'two years' for all species but nothing can ensure us that this is a reasonable choice. We have now discussed this point and specified that some uncertainty surrounds our results on how fast the predicted changes of biodiversity may occur and how quick the associated conservation planning we recommend should be applied (L298-304). As global change and the current 5th biodiversity extinction, although we know which negative environmental transformations are to be expected, the rate of those always remains highly uncertain.

3-2) In addition, the authors considered dispersal scenarios for plants over European Alps but imposed a no-dispersal rule for all species at low elevations from surrounding regions. The omission of plant colonization from other regions would lead to an overestimation of the biodiversity loss, and undermine the validity of main findings. I would suggest to explore and discuss the inclusion of species dispersion from surrounding regions at low elevations in shaping the future plant biodiversity.

You are entirely right. This an interesting aspect that has not been discussed due to word limitation in another submission. We have therefore discussed this important point in the last (added) section related to challenges. On the one hand, we explained that due to observational and dispersal data limitation, it was hard to consider these outside migrations which could increase species diversity at low elevation. On the other hand, we highlight that, due to the absence of competition in the models, it is also hard to predict how such migrations would impact the future diversity of the Alps. As in this work, migration (+competition) could have instead a negative impact on the future biodiversity of the Alps. Also considering no plant dispersal from the outside is very informative as it gives an objective overview

of how the sole original biodiversity of the European Alps would evolve according to future projections (the known 4'250 inventoried species).

3-3) Second, the authors discussed the impact of land cover change on future plant biodiversity changes, but there is no actual data on this effect. Previous research has demonstrated an important impact of historical land cover changes on biodiversity changes (Garcés-Pastor et al., 2022). It is therefore important to compare the direction and rate of biodiversity changes due to land cover changes at least from the historical perspective with future climate change-induced biodiversity changes over the European Alps. Furthermore, there is clear evidence to show that the warming-induced upward shift of the life-form tree would compress the physical habitat area of alpine plants (Körner, 2012; Greenwood & Jump, 2018), which could potentially affect the species rarity as calculated in this study. The omission of tree upslope might induce an increase instead of a decline in endemism at the taxonomic level as shown in Figure 1.

Thank you for your comment. While we included information on %land cover change (categorical) between current and future, as well as their respective levels of diversity over the study region (Fig. S5-6), we forgot to link both aspects by including an analysis on how diversity changes between current land cover types and their future transformation. Within pixels with changing land cover, we found an overall decreasing trends between current and future conditions for all land cover types. We included the results in Fig. S7 for both land cover models (by 2050, for SSP245) and added substantial text to the discussion including relevant additional comments and references that were here given (L-109-120).

3-4) Third, to achieve the biodiversity protection objectives of COP15, this study has increased the coverage of protected areas in the Alps to approximately 30%. However, it is still unclear how effective these protected areas are. By identifying species that are not currently covered by protected areas, targeted strategies can be developed to address these gaps and improve overall biodiversity conservation outcomes. Therefore, conducting a systematic assessment of gap species is crucial for effective biodiversity protection, especially for threatened and restricted range species (Rodrigues et al., 2004).

Although some species ranges assessments between current and future PAs were done (Fig. S18a-d), it is true we forgot to assess the quality of the predicted reserve networks. We have therefore added additional GAP analyses (Fig. S9) that shows how the new SCP PAs better protect multifaceted diversity

than IUCN I-II and IUCN I-VI. Discussion on this point was also added (L249-280). Thank you for your input that has improved the manuscript and the discussion.

3-5) Lastly, it is expected that in a warmer climate, plants will migrate towards colder regions encompassing both high elevations and latitudes. A comprehensive understanding of future plant migration can be achieved by analysing the two-dimensional changes in elevation and latitude together rather than separately. Conducting separate analyses across different sections may lead to fragmented knowledge about local conservation efforts. The authors suggest that optimizing conservation efforts in the Alps may require little European coordination; however, this contradicts the fact that plant migration is expected to occur across various regions and countries.

Thank you for your suggestion. We have added extra-analyses on latitudinal and elevational dispersal in Fig. S3ab, which nicely complement our previous results by showing that most of the plant (elevation) classes migrates indeed towards higher elevation and latitude.

Minor comments:

3-6) 1. Line 447: Please include sensitivity tests for various parameters utilized in simulating seed dispersal, such as `dispKernel`, barrier, and others.

Related to comment **(3-1)**. We have now added a discussion on this part.

3-7) 2. In line 473, the authors describe a technique they developed to combine various seed dispersal scenarios. This technique incorporates 13 different scenarios that can be grouped into three categories: realistic, unrestricted, and no dispersal. Please add details and reasoning behind each step.

We have fully modified the *Projection* section. This has therefore been made clearer.

3-8) 3. The results shown in Figure 1 demonstrate a significant variation in the endemism of alpine flora across functional, taxonomic, and phylogenetic dimensions. What are the underlying causes of this pattern?

As initially explained in the text (L105-108), the increasing endemism and rarity at higher elevation is mainly due to the loss of range size for ~70% of the species. Species are losing their ranges because, when migrating to higher elevation, they suffer from dispersal lags (local extinctions due to insufficient dispersal capacities/rates compared to that of global change) and decreasing physical habitats (the higher they go, the less surface they have, and therefore less habitat). Maybe we were not clear enough on this part. We have therefore added a sentence (L142-143) , but also (related to your comment **3-3**) added a small discussion on how increasing endemism and rarity could generally be caused by increasing forest successions and expansions due to land cover change and increased warming (L118-120).Our References

- Bellamy, C. et al. 2013. Multiscale, presence-only habitat suitability models: fine-resolution maps for eight bat species. - *Journal of Applied Ecology* 50: 892–901.
- Bonnet-Lebrun, A. S. et al. 2020. Identifying priority conservation areas for a recovering brown bear population in Greece using citizen science data. - *Animal Conservation* 23: 83–93.
- Botella, C. et al. 2020. Bias in presence-only niche models related to sampling effort and species niches: Lessons for background point selection. - *PLoS ONE* 15: 1–18.
- Carlson, C. J. et al. 2022. Climate change increases cross-species viral transmission risk. - *Nature* 607: 555–562.
- Chauvier, Y. et al. 2021. Novel methods to correct for observer and sampling bias in presence-only species distribution models. - *Global Ecology and Biogeography* 30: 2312–2325.
- Chauvier, Y. et al. 2022. Resolution in species distribution models shapes spatial patterns of plant multifaceted diversity. - *Ecography* 2022: e05973.
- Connor, T. et al. 2018. Effects of grain size and niche breadth on species distribution modeling. - *Ecography* 41: 1270–1282.
- El-Gabbas, A. and Dormann, C. F. 2018. Improved species-occurrence predictions in data-poor regions: using large-scale data and bias correction with down-weighted Poisson regression and Maxent. - *Ecography* 41: 1161–1172.
- Gaüzère, P. et al. 2022. The diversity of biotic interactions complements functional and phylogenetic facets of biodiversity. - *Current Biology* 32: 2093–2100.e3.
- Gray, C. L. et al. 2016. Local biodiversity is higher inside than outside terrestrial protected areas worldwide. - *Nature Communications in press*.
- Hoffmann, S. et al. 2018. Uniqueness of protected areas for conservation strategies in the European Union. - *Scientific Reports* 8: 1–14.
- Kujala, H. et al. 2013. Conservation Planning with Uncertain Climate Change Projections. - *PLoS ONE in press*.
- Kukkala, A. S. et al. 2016. Matches and mismatches between national and EU-wide priorities: Examining the Natura 2000 network in vertebrate species conservation. - *Biological Conservation* 198: 193–201.

- Merow, C. et al. 2017. Integrating occurrence data and expert maps for improved species range predictions. - *Global Ecology and Biogeography* 26: 243–258.
- Mertes, K. and Jetz, W. 2018. Disentangling scale dependencies in species environmental niches and distributions. - *Ecography* 41: 1604–1615.
- O'Connor, L. M. J. et al. 2021. Balancing conservation priorities for nature and for people in Europe. - *Science* 373: 856–860.
- Orlikowska, E. H. et al. 2016. Gaps in ecological research on the world's largest internationally coordinated network of protected areas: A review of Natura 2000. - *Biological Conservation* 200: 216–227.
- Pollock, L. J. et al. 2017. Large conservation gains possible for global biodiversity facets. - *Nature* 546: 141–144.
- Pollock, L. J. et al. 2020. Protecting Biodiversity (in All Its Complexity): New Models and Methods. - *Trends in Ecology and Evolution* 35: 1119–1128.
- Renner, I. W. and Warton, D. I. 2013. Equivalence of MAXENT and Poisson Point Process Models for Species Distribution Modeling in Ecology. - *Biometrics* 69: 274–281.
- Renner, I. W. et al. 2015. Point process models for presence-only analysis. - *Methods in Ecology and Evolution* 6: 366–379.
- Renner, I. W. et al. 2019. Combining multiple data sources in species distribution models while accounting for spatial dependence and overfitting with combined penalized likelihood maximization. - *Methods in Ecology and Evolution* 10: 2118–2128.
- Thuiller, W. et al. 2015. From species distributions to meta-communities. - *Ecology Letters* 18: 1321–1328.
- Thuiller, W. et al. 2018. Combining point-process and landscape vegetation models to predict large herbivore distributions in space and time—A case study of *Rupicapra rupicapra*. - *Diversity and Distributions* 24: 352–362.
- Warton, D. I. and Shepherd, L. C. 2010. Poisson point process models solve the “pseudo-absence problem” for presence-only data in ecology. - *Annals of Applied Statistics* 4: 1383–1402.
- Watson, J. E. M. et al. 2014. The performance and potential of protected areas. - *Nature* 515: 67–73.

Decision Letter, first revision:

20th August 2023

Dear Dr Chauvier-Mendes,

Your manuscript entitled "Transnational conservation to anticipate future plant shifts in Europe" has now been seen by our three original reviewers, whose comments are attached. The reviewers are encouraged by the revisions, but have raised a number of concerns which will need to be addressed before we can offer publication in Nature Ecology & Evolution. We will therefore need to see your responses to these remaining concerns, along with a revised manuscript, before we can reach a final decision regarding publication. We may be able to form a decision on the revision without sending it back to any of the reviewers, but we will decide that when we see the revision and the point-by-point response.

As you prepare your revision, please note that there is no word limit to the Methods section of the main file, and that the main text (Introduction, Results and Discussion) can be up to 3500 words. Please also note that Extended Data figures can be included as well as, or instead of, Supplementary Figures. You may choose to change some of your current Supplementary Information to Extended Data figures (these appear in-line with the main text online).

We therefore invite you to revise your manuscript taking into account all reviewer and editor comments. Please highlight all changes in the manuscript text file.

- * Include a "Response to reviewers" document detailing, point-by-point, how you addressed each reviewer comment. If no action was taken to address a point, you must provide a compelling argument. This response will be sent back to the reviewers along with the revised manuscript.
- * If you have not done so already please begin to revise your manuscript so that it conforms to our Article format instructions at <http://www.nature.com/natecolevol/info/final-submission>. Refer also to any guidelines provided in this letter.
- * Include a revised version of any required reporting checklist. It will be available to reviewers to aid

34in their evaluation if the manuscript goes back for peer review. A revised checklist is essential for re-review of the paper.

[REDACTED]

[REDACTED]

Reviewers' comments:

Reviewer #1 (Remarks to the Author):

Thank you very much for the opportunity to review manuscript NATECOLEVOL-23030689A, "Transnational conservation to anticipate future plant shifts in Europe". I really appreciate your thoughtful responses to reviewer comments and the modifications you have made to your work. In my opinion this is a much improved manuscript now and will make it easier for others to understand your work.

I do only have two small comments at this point, only the second of which needing attention.

First, the benefits of Zonation are slightly overstated, but this is such a small thing I think that its not really worth going into much detail or modifying the text further. I think the rational for using Zonation now makes a lot of sense.

Second, Data and materials availability. Following the link, I can see the EnviDat repository, but this time the link just shows an error that the metadata is missing and I can't even see any files in there.

Best regards,

35Richard Schuster, PhD

Director of Spatial Planning and Innovation, Nature Conservancy of Canada
Email: richard.schuster@natureconservancy.ca

Reviewer #2 (Remarks to the Author):

Thank you for the opportunity to re-review the manuscript "Transnational conservation to anticipate future plant shifts in Europe". The authors have provided many revisions in response to my and the other reviewers' earlier concerns. In general, their responses and revisions have adequately addressed the main issues that I had raised, but I still found the study difficult to follow and that the manuscript lacked clarity in several places.

Most importantly, I still felt that the main text lacked enough sufficient methodological detail in many places that I think would result in a large proportion of readers struggling to understand the authors' approach and thus trust the overall findings of the study.

The new Fig. S1 goes a long way to addressing this issue by providing a schematic overview of the study design. While there are some issues with this figure that I elaborate on below, in general I think it deserves to be refined and moved to the main text as a crucial aid to readers.

Alongside this, I suggest that the authors do a bit more to add important and necessary methodological details in several of the sections in the main text. Please see below for where I think more may be warranted.

In addition, I still have some issues with the conclusions reached based on the analysis focusing on only IUCN I-II and Nature 2000 PAs. I understand and am OK with the rationale for only considering these in the prioritization exercise, but I feel that it is important to acknowledge the presence of the many other PAs in the landscape that may align with the analysis' resulting priorities and what role they might play in achieving a robust conservation network under 30x30, e.g., if they strengthened or altered management in some way to be more biodiversity focused. As the paper currently reads, these PAs are completely disregarded and following the priorities and recommendations of this paper would imply that far greater than 30% of the Alps' area would need to be under protection, which does not seem practical and ignores potentially established management and conservation operations in many places.

Finally, the inclusion of the caveats and challenges section is very appreciated, but I didn't see a strong response to my original comment as to the need to include a more robust conclusion to the paper. The authors still have not put the findings and conclusions of this study into the context of the many other regional and global conservation prioritizations out there that focus on different types of biodiversity and different conservation objectives that are all very relevant for 30x30 planning. I think a large proportion of readers, as well as the target audience and general stakeholders in this area would be interested in knowing how the locations of priorities from this study may be aligned with

36other priorities, and where there might be important tradeoffs to consider.

Overall, I'm satisfied with the methodological approaches, and I feel that the issues raised above can likely be addressed with further revision to lead to an improved and more accessible paper.

I provide further comments below:

Abstract – I would replace the word “outdated” with something else that more explicitly states what you mean. For the same sentence, you should provide more detail as to how your conservation simulations are improvements (also, your simulations don't “protect” biodiversity, they might “capture” biodiversity patterns, but they do not represent actual protected areas).

Upward shifts of multifaceted diversity – first paragraph, second sentence: change to “...several carbon emissions and dispersal scenarios...”.

Upward shifts of multifaceted diversity – second paragraph: are species at high elevations going locally extinct though following upward range shifts, even if overall diversity is increasing? How would this affect the timing of protected areas needs at high elevations?

Species turnover and extinctions – first paragraph: here you say many lowland species are forecast to experience strong range expansion, but in the previous section you say that most loss is expected to occur primarily at low elevations. Which is it? Are low elevation species losing species multifaceted diversity but gaining in distribution? And high elevation species are gaining in species multifaceted diversity, but losing in distribution? Is this an issue for determining conservation priorities given that the prioritization only focuses on diversity metrics?

Species turnover and extinctions – last paragraph, last sentence: which species would be expected to migrate from high to low elevations, if most species are following upward shifts? Also, couldn't species migrating upslope face competitive exclusion from species already established at higher elevations?

Local conservation prioritization – the level of detail presented on the ABF algorithm in text here should be expanded; I don't think it's enough to say what the primary objective is, the ABF algorithm should be explained in a basic and understandable way here (the acronym is not even provided until the methods 11 pages later).

Regional conservation prioritization – similar to the above comment, more detail is needed on the CAZ algorithm here to aid in understanding this approach.

Also, the last sentence of the first paragraph is difficult to follow. What does it mean for regional conservation needs to be similar to local ones but be more “distinct”? And what do you mean by “protect as efficiently multifaceted diversity”?

I'm not sure I follow the last sentence of the second paragraph of this section either. What do you mean by “strong spatial and political divergence in optimized PA expansion”? Do you mean that priorities are sporadically spread across the Alps thus leading to differences in the degree to which

37different countries would need to add protection?

The final paragraph gets at this a bit by discussing the “ideal” conservation contributions for different countries, but it isn’t clear to what degree the contributions for protection at different elevations is a function of the availability of land area at different elevations versus the protection of those different elevational strata by country. For example, the text says that Switzerland should expand across all strata, Austria should focus on mid-elevations, and France and Germany should focus on low elevations. Is this the case simply because those countries have more available land area to protect at these elevations (e.g., France and Germany have more lowland area than Switzerland, so they have a higher burden for lowland conservation), or are conservation priorities truly disproportionately high in these elevation strata when accounting for the available area?

EU conservation perspectives

Choice of IUCN I-II PAs – I appreciate the information provided in the response and feel, given the EU context, the choice of using I-II PAs is acceptable. However, I think it might still be important to highlight in some way where III-VI PAs might be included in the prioritization solution as these areas may already have some provisions for biodiversity conservation and perhaps improvements to management or stricter management plans would only be necessary to achieve biodiversity conservation goals, whereas areas that are included in the solution with no form of protection would likely need far more inputs (e.g., restoration, management, governance, rights resolutions, infrastructure, etc.). Perhaps including a map like the inset map of Fig. 4A but showing where different scenarios overlap with less strict (III+) PAs could be a useful addition.

I feel that advocating for excluding III-VI from any sort of consideration is not wholly justified, and furthermore that in practice it seems unlikely that management and protection of areas under III-VI PAs will not simply cease with the expansion of PAs to cover the areas proposed by this study. Thus, in reality, the PA network would cover a far greater area than ~35%, which may not be realistic given other land use needs. It seems more prudent to advocate for alternate management and/or restoration in areas that have some level of protection and that could help to build a more resilient PA network if “upgraded” to focus more explicitly on multifaceted conservation.

In the final paragraph, I think it would be more accurate to state that Switzerland is predicted to bear the brunt of the effort in expanding the transnational network in the Alps not only because of a lack of PAs, but because of the high multifaceted biodiversity value in the country. If the country had low biodiversity value, no areas would show up as important for PA expansion, so it’s not just about the lack of PAs currently.

Challenges

I appreciate the added text to the challenges. One recommendation would be to highlight how the results of this paper support or contract findings from the other recent EU-focused prioritizations mentioned that include other aspects of biodiversity and ecosystem services. If there are synergies or tradeoffs with proposed PA expansions from those works (or from other, global studies), it would be helpful to know.

38Figures and Tables

Fig. 3 – I’m still a little confused by this plot, mainly as to how to interpret the areas of convergence. If the grey areas in the map are indicating priorities in both current and future time periods, then how do you distinguish portions of the grey space that are perfectly between current and 2080, or perfectly between current and 2050? Also, I’m not sure where the PAs are in this map, I don’t see any legible contours as described in the caption.

Fig. 4 – In the right map of panel A, I don’t see a lot of priorities for regional expansion showing up that would align with the ‘conservation convergence’ areas shown in Fig. 3, though I would expect to. If one of the messages from the local and regional conservation prioritization sections is that PA expansion is needed to connect different elevational strata and also bolster latitudinal connectivity, then I would have expected more areas showing up in the 3-time overlap around the conservation convergence areas highlighted in Fig. 3.

Also, since one of the main links to this figure from the text concerned the need for transnational conservation, it might help to add the political boundaries to the right map in panel A and to highlight where truly transnational expansion is necessary, versus where individual countries need to (independently) expand their PA networks.

To make panel B clearer, it might help to just have the five sets of bar charts in a single row with titles indicating the elevation strata. The mountain graphic is already shown and described in Fig. 2 so is not needed here. Removing it would make the figure less busy and would allow you to make the bar charts bigger and the text more legible. Another option would be to move the right map of panel A underneath the elevation map of panel A, then make the bar charts a vertical set of five panels labelled by strata, which would better mimic their orientation along an elevation gradient. Either option would take up about the same amount of space on the page.

Fig S1 – this schematic workflow is a helpful addition to orient the reader to the methodology, but I find it a bit confusing. For example, it doesn’t explain why bias correction was only applied for a subset of the species, and it’s not clear what an “intensity map” is. I was also confused a bit by how the four future scenarios were constructed, given that these should be from all possible combinations of climate models (ensembled), land cover models, emissions scenarios, and time horizons. I would expect this to be 6 futures, even if the VOLANTES-HERCULES is available for only 2041-2060:

Climate ensemble + ALARM-ECOCHANGE + SSP245 + 2041-2060
Climate ensemble + ALARM-ECOCHANGE + SSP585 + 2041-2060
Climate ensemble + ALARM-ECOCHANGE + SSP245 + 2071-2090
Climate ensemble + ALARM-ECOCHANGE + SSP585 + 2071-2090
Climate ensemble + VOLANTES-HERCULES + SSP245 + 2041-2060
Climate ensemble + VOLANTES-HERCULES + SSP585 + 2041-2060

Unless the authors are using an ensemble of the land cover models too?

Also, there are many acronyms in this schematic that aren't explained or defined (TSS, SCP, CAZ, ABF, TD, rPD, rFD, WE, rPE, rFE, PR, FR).

Given that a clear understanding of the results depends in large part on following the complex and multifaceted analysis that is summarized in this schematic, I suggest the authors refine this figure and move it into the main text (I believe there is room for an additional figure), including defining all key terms and orienting readers to the different aspects (phases) of the analysis, from data compilation/filtering/correction, to species modeling (including variants with different dispersal parameters), creating different types of diversity maps, and conducting the prioritization. The climate, land cover, emissions scenarios, and time horizons could be included here as they are, or included as a separate table.

Table S2 – I note in your response that the acronyms for data sources are described, but I don't see where. It would be helpful to point the reader to that place somewhere in Table S2 (in the caption, or as a note below the table).

Methods

Climate data – CHELSA CMIP6 climatologies are available for 30 year time slices; for the future periods, should this read 2041-2070 and 2071-2100, in line with what is available at the link provided? Also, only 5 GCMs are available at that website, so it would be helpful to note how MIROC6, AWI-CM-1-1-MR, EC-Earth3, INM-CM5-0 were obtained (or how others wanting to reproduce this analysis can access these data).

Systematic conservation planning, Zonation – Some of the detail in this section should be included in the main text. In particular, details from the last paragraph that discuss the number of scenarios, the included parameters, and the various Zonation algorithms I think are central enough to the paper to warrant inclusion in the main text in the local and regional conservation prioritization sections.

Systematic conservation planning, post-analyses – I'm not sure I follow why the top 20% was chosen. Is this because currently only 10% of the study area is included in IUCN I-II and Natura 2000 sites, and an additional 20% would bring the total to 30% in line with Target 3 of the GBF?

Reviewer #3 (Remarks to the Author):

I appreciate a lot the authors' efforts in revising the manuscript with my previous comments. The manuscript has been improved to a large extent regarding the description of parameters used, the discussions on uncertainties associated with the land cover change, and associated with latitudinal and elevational dispersal. However, I do have some reservations regarding the lack of quantitative uncertainty assessment associated with selection of a few key parameter values.

While I agree with the authors that it is not realistic to make all key parameters species specific, the current treatment of these parameter values is not informative to readers. I suggest the authors to

40quantitatively evaluate the robustness of their results based on sensitivity tests to parameter values within the reasonable range.

*****END*****

Author Rebuttal, first revision:

Reviewer #1 (Remarks to the Author):

(1) Thank you very much for the opportunity to review manuscript NATECOLEVOL-23030689A, “Transnational conservation to anticipate future plant shifts in Europe”. I really appreciate your thoughtful responses to reviewer comments and the modifications you have made to your work. In my opinion this is a much improved manuscript now and will make it easier for others to understand your work.

I do only have two small comments at this point, only the second of which needing attention.

First, the benefits of Zonation are slightly overstated, but this is such a small thing I think that its not really worth going into much detail or modifying the text further. I think the rational for using Zonation now makes a lot of sense.

Second, Data and materials availability. Following the link, I can see the EnviDat repository, but this time the link just shows an error that the metadata is missing and I can't even see any files in there.

Thank you for your positive feedbacks, this is very encouraging. Regarding data accessibility, this is very odd, as we have solved the problem with people from Envidat. We have tried again to access the data from different computers using Cyberduck and everything is working fine. It is possible that you tried to access the servers while updates from the data portal were ongoing. Please make sure to tell us if the problem arises again...

Reviewer #2 (Remarks to the Author):

(2) Thank you for the opportunity to re-review the manuscript “Transnational conservation to anticipate future plant shifts in Europe”. The authors have provided many revisions in response to my and the other reviewers' earlier concerns. In general, their responses and revisions have adequately addressed

41the main issues that I had raised, but I still found the study difficult to follow and that the manuscript lacked clarity in several places.

Most importantly, I still felt that the main text lacked enough sufficient methodological detail in many places that I think would result in a large proportion of readers struggling to understand the authors' approach and thus trust the overall findings of the study.

The new Fig. S1 goes a long way to addressing this issue by providing a schematic overview of the study design. While there are some issues with this figure that I elaborate on below, in general I think it deserves to be refined and moved to the main text as a crucial aid to readers.

Alongside this, I suggest that the authors do a bit more to add important and necessary methodological details in several of the sections in the main text. Please see below for where I think more may be warranted.

In addition, I still have some issues with the conclusions reached based on the analysis focusing on only IUCN I-II and Nature 2000 PAs. I understand and am OK with the rationale for only considering these in the prioritization exercise, but I feel that it is important to acknowledge the presence of the many other PAs in the landscape that may align with the analysis' resulting priorities and what role they might play in achieving a robust conservation network under 30x30, e.g., if they strengthened or altered management in some way to be more biodiversity focused. As the paper currently reads, these PAs are completely disregarded and following the priorities and recommendations of this paper would imply that far greater than 30% of the Alps' area would need to be under protection, which does not seem practical and ignores potentially established management and conservation operations in many places.

Thank you for your feedback. Please find our responses to these points below.

(3) Finally, the inclusion of the caveats and challenges section is very appreciated, but I didn't see a strong response to my original comment as to the need to include a more robust conclusion to the paper. The authors still have not put the findings and conclusions of this study into the context of the many other regional and global conservation prioritizations out there that focus on different types of biodiversity and different conservation objectives that are all very relevant for 30x30 planning. I think a large proportion of readers, as well as the target audience and general stakeholders in this area would be interested in knowing how the locations of priorities from this study may be aligned with other priorities, and where there might be important tradeoffs to consider.

We have to admit that we have a problem understanding what the suggestion here is. Putting our findings and conclusions into the context of other conservation prioritizations implies accessing other regional studies that cover the European Alps (including the data). This is the first SCP study over this mountain region; therefore, comparing our locations of conservation priorities to that of other studies seems complicated. Some European and global conservation prioritizations have been of course published; however, the spatial and taxonomic precision/extent of these studies does not allow a fair visual comparison, especially when Switzerland is always missing at the European level (Allan et al., 2022; Brum et al., 2017; Jung et al., 2021; McGowan et al., 2020; O'Connor et al., 2021; Pollock et al., 2017; Soto-Navarro et al., 2020; Vitt et al., 2023).

(4) Overall, I'm satisfied with the methodological approaches, and I feel that the issues raised above can likely be addressed with further revision to lead to an improved and more accessible paper.

I provide further comments below:

Abstract – I would replace the word “outdated” with something else that more explicitly states what you mean. For the same sentence, you should provide more detail as to how your conservation simulations are improvements (also, your simulations don't “protect” biodiversity, they might “capture” biodiversity patterns, but they do not represent actual protected areas).

We have changed the sentence.

(5) Upward shifts of multifaceted diversity – first paragraph, second sentence: change to “...several carbon emissions and dispersal scenarios...”.

We have included this additional word.

(6) Upward shifts of multifaceted diversity – second paragraph: are species at high elevations going locally extinct though following upward range shifts, even if overall diversity is increasing? How would this affect the timing of protected areas needs at high elevations?

While the first section explains that diversity at high elevation increases due to upward migration of species ranges, the second section highlights that, despite these diversity changes, species are also expected to lose their range for two reasons: (1) range losses due to habitat limitation (species migrating higher have less physical space available) and (2) unsuitable environmental conditions leading to local extinctions (lags between species migration and environmental changes). We have changed the sentence to improve clarity (L141-146). Regarding the timing, as many predictions and forecasts, we do not know how fast would undergo species migration. We therefore recommend by 2030 the

43establishment of the improved EU network to assist as soon as possible species and biodiversity migrations (L313-315).

(7) Species turnover and extinctions – first paragraph: here you say many lowland species are forecast to experience strong range expansion, but in the previous section you say that most loss is expected to occur primarily at low elevations. Which is it? Are low elevation species losing species multifaceted diversity but gaining in distribution? And high elevation species are gaining in species multifaceted diversity, but losing in distribution? Is this an issue for determining conservation priorities given that the prioritization only focuses on diversity metrics?

We think there is a confusion here. Diversity at low elevation decreases because 70% of the species (from all elevation strata) loses their range (Fig. 2) due to migration towards higher elevation (Fig. 3ab). The other 30% (most of it being lowland species) gain ranges, but it does not mean that it is enough to compensate the observed diversity losses we found in the lowlands. We added bit of text L136 to clarify. Regarding conservation, as explain in the *Zonation* section, prioritization does not only focus on diversity metrics per pixel, but also on features (singular species distribution and how unique communities may be across regions).

(8) Species turnover and extinctions – last paragraph, last sentence: which species would be expected to migrate from high to low elevations, if most species are following upward shifts? Also, couldn't species migrating upslope face competitive exclusion from species already established at higher elevations?

It is generally known that competition at low elevations restrict species from higher elevations from moving down the gradient (Choler et al., 2001). Therefore, in the event of upward migrations, species already established at higher elevations will be the ones facing competitive exclusion from species migrating upslope (Alexander et al., 2015). Since our models cannot account for this effect, it appears that unrealistic downwards migrations were found in our results. But it is interesting to note that in theory, species from higher elevation that are more “generalists” (e.g., *Saxifraga oppositifolia*), could migrate to lower elevation in the absence of competition. We have reformulated the sentence to avoid confusion (L157-159).

(9) Local conservation prioritization – the level of detail presented on the ABF algorithm in text here should be expanded; I don't think it's enough to say what the primary objective is, the ABF algorithm should be explained in a basic and understandable way here (the acronym is not even provided until the methods 11 pages later).

We are confused as a clear revision was added to the text (L171-175): *“Local prioritisation (considering alpha/pixel biodiversity) is a strategy that focuses more on protecting given localities and areas that are biodiversity rich within a given region, i.e., local hotspots. For this, we employed the ABF algorithm of the Zonation software (see Zonation section in Methods) with our species model outputs and their phylogenetic/functional information”*. Same goes for CAZ (L207-213): *“Unlike local optimisation, regional conservation prioritisation (maximizing gamma/regional biodiversity) does not necessarily focus on single multifaceted-rich localities but rather on unique localities and their complementarity, i.e., magnifying the multifaceted diversity of the whole region. For this, we employed the CAZ (core-area Zonation) algorithm of the Zonation software (see Zonation section in Methods), and evaluated how could regional conservation strategies adapt to multifaceted diversity changes in the European Alps and improve its protection”*. We think that being more specific about the methods in the main text would on the contrary impend the readability of the manuscript. We have however defined in the main text the acronym ABF (additive benefit function) and CAZ (core-area zonation).

(10) Regional conservation prioritization – similar to the above comment, more detail is needed on the CAZ algorithm here to aid in understanding this approach.

See our answer above. Thank you.

(11) Also, the last sentence of the first paragraph is difficult to follow. What does it mean for regional conservation needs to be similar to local ones but be more “distinct”? And what do you mean by “protect as efficiently multifaceted diversity”?

We have clarified the sentence (L213-216).

(12) I’m not sure I follow the last sentence of the second paragraph of this section either. What do you mean by “strong spatial and political divergence in optimized PA expansion”? Do you mean that priorities are sporadically spread across the Alps thus leading to differences in the degree to which different countries would need to add protection?

We understand the confusion. We have therefore improved this part of the text (L238-242), thank you.

(13) The final paragraph gets at this a bit by discussing the “ideal” conservation contributions for different countries, but it isn’t clear to what degree the contributions for protection at different elevations is a function of the availability of land area at different elevations versus the protection of those different elevational strata by country. For example, the text says that Switzerland should expand across all strata, Austria should focus on mid-elevations, and France and Germany should focus on low

elevations. Is this the case simply because those countries have more available land area to protect at these elevations (e.g., France and Germany have more lowland area than Switzerland, so they have a higher burden for lowland conservation), or are conservation priorities truly disproportionately high in these elevation strata when accounting for the available area?

We are not sure to fully understand the question. We have modified the legend in figure 4 and removed “(% area)” to avoid confusion. The percentage of PAs per political unit in figure 4 is based on the current distribution of PAs and the expected ones according to our conservation simulations. These distributions are independent from the distribution of available areas across elevation and might change from countries to countries. PA recommendations per countries are actually partly driven by the lack of current protection for some elevation strata (e.g. all strata for Switzerland, lower strata for Italy or Austria). However, looking at the map, we can easily decipher and understand why no conservation recommendations pop up at high elevation for Germany and Slovenia. These two countries do neither possess Alpine nor Nival ecosystems.

(14) EU conservation perspectives

Choice of IUCN I-II PAs – I appreciate the information provided in the response and feel, given the EU context, the choice of using I-II PAs is acceptable. However, I think it might still be important to highlight in some way where III-VI PAs might be included in the prioritization solution as these areas may already have some provisions for biodiversity conservation and perhaps improvements to management or stricter management plans would only be necessary to achieve biodiversity conservation goals, whereas areas that are included in the solution with no form of protection would likely need far more inputs (e.g., restoration, management, governance, rights resolutions, infrastructure, etc.). Perhaps including a map like the inset map of Fig. 4A but showing where different scenarios overlap with less strict (III+) PAs could be a useful addition.

I feel that advocating for excluding III-VI from any sort of consideration is not wholly justified, and furthermore that in practice it seems unlikely that management and protection of areas under III-VI PAs will not simply cease with the expansion of PAs to cover the areas proposed by this study. Thus, in reality, the PA network would cover a far greater area than ~35%, which may not be realistic given other land use needs. It seems more prudent to advocate for alternate management and/or restoration in areas that have some level of protection and that could help to build a more resilient PA network if “upgraded” to focus more explicitly on multifaceted conservation.

Thank you very much, this is indeed a nice addition. We have added a figure in the appendix (S20) and readapt the text L268-275.

46(15) In the final paragraph, I think it would be more accurate to state that Switzerland is predicted to bear the brunt of the effort in expanding the transnational network in the Alps not only because of a lack of PAs, but because of the high multifaceted biodiversity value in the country. If the country had low biodiversity value, no areas would show up as important for PA expansion, so it's not just about the lack of PAs currently.

You are indeed right, and we have therefore improved the sentence (L282-283).

(16) Challenges

I appreciate the added text to the challenges. One recommendation would be to highlight how the results of this paper support or contract findings from the other recent EU-focused prioritizations mentioned that include other aspects of biodiversity and ecosystem services. If there are synergies or tradeoffs with proposed PA expansions from those works (or from other, global studies), it would be helpful to know.

See our answer above in comment (3). Thank you.

(17) Figures and Tables

Fig. 3 – I'm still a little confused by this plot, mainly as to how to interpret the areas of convergence. If the grey areas in the map are indicating priorities in both current and future time periods, then how do you distinguish portions of the grey space that are perfectly between current and 2080, or perfectly between current and 2050? Also, I'm not sure where the PAs are in this map, I don't see any legible contours as described in the caption.

We understand the confusion. The map is constructed based on RGB colours. According to this method, more grey areas normally describe a convergence in the values of the three layers (so here, current, 2050 and 2080). Here the most important convergences are of course where IUCN I-II + natura 2000 network are located (they were included in in the expansion SCPs). Therefore we changed the legend (grey with contour = PAs) and focus the figure description on the visible colours for less confusion.

(18) Fig. 4 – In the right map of panel A, I don't see a lot of priorities for regional expansion showing up that would align with the 'conservation convergence' areas shown in Fig. 3, though I would expect to. If one of the messages from the local and regional conservation prioritization sections is that PA expansion is needed to connect different elevational strata and also bolster latitudinal connectivity, then I would

47have expected more areas showing up in the 3-time overlap around the conservation convergence areas highlighted in Fig. 3.

This comment is related to above. Deciphering these overlaps using an RGB method is not easy when the three layers have a common PAs network as “priority areas” (unlike Fig. 16ab). RGB normally shows greyer colour for overlaps but we agree we do not see them as well as the PAs. We therefore decided to focus the description of fig. 3 on the visible “hotspots” colours, and fig. 4 on the visible conservation overlaps.

(19) Also, since one of the main links to this figure from the text concerned the need for transnational conservation, it might help to add the political boundaries to the right map in panel A and to highlight where truly transnational expansion is necessary, versus where individual countries need to (independently) expand their PA networks.

The political borders added on the left panel + information on elevation was to actually run this comparison. We are afraid that adding the political borders on the right panel would on the contrary impends the readability of the figure.

(20) To make panel B clearer, it might help to just have the five sets of bar charts in a single row with titles indicating the elevation strata. The mountain graphic is already shown and described in Fig. 2 so is not needed here. Removing it would make the figure less busy and would allow you to make the bar charts bigger and the text more legible. Another option would be to move the right map of panel A underneath the elevation map of panel A, then make the bar charts a vertical set of five panels labelled by strata, which would better mimic their orientation along an elevation gradient. Either option would take up about the same amount of space on the page.

Thank you very much for the suggestion, we have adapted a new figure mixing the second idea with a new mountain and ecosystems representation. Figure S17a was therefore also changed.

(21) Fig S1 – this schematic workflow is a helpful addition to orient the reader to the methodology, but I find it a bit confusing. For example, it doesn’t explain why bias correction was only applied for a subset of the species, and it’s not clear what an “intensity map” is. I was also confused a bit by how the four future scenarios were constructed, given that these should be from all possible combinations of climate models (ensembled), land cover models, emissions scenarios, and time horizons. I would expect this to be 6 futures, even if the VOLANTES-HERCULES is available for only 2041-2060:

Climate ensemble + ALARM-ECOCHANGE + SSP245 + 2041-2060

Climate	ensemble	+	ALARM-ECOCHANGE	+	SSP585	+	2041-2060
Climate	ensemble	+	ALARM-ECOCHANGE	+	SSP245	+	2071-2090
Climate	ensemble	+	ALARM-ECOCHANGE	+	SSP585	+	2071-2090
Climate	ensemble	+	VOLANTES-HERCULES	+	SSP245	+	2041-2060
Climate	ensemble	+	VOLANTES-HERCULES	+	SSP585	+	2041-2060

Unless the authors are using an ensemble of the land cover models too?

Also, there are many acronyms in this schematic that aren't explained or defined (TSS, SCP, CAZ, ABF, TD, rPD, rFD, WE, rPE, rFE, PR, FR).

Thank you for your comment. We have replaced in Fig. S1 "Intensity maps" by "SDM maps" and added a reference to the EBC section which now explains that EBC only applied to 1'248 species that showed environmental bias within their observations (L445-446). As for the future scenarios, land cover was indeed also ensembled, hence the four averaged future scenarios. We have specified it in the figure and also in the main text L484. We have also described the acronyms in the legend.

(22) Given that a clear understanding of the results depends in large part on following the complex and multifaceted analysis that is summarized in this schematic, I suggest the authors refine this figure and move it into the main text (I believe there is room for an additional figure), including defining all key terms and orienting readers to the different aspects (phases) of the analysis, from data compilation/filtering/correction, to species modeling (including variants with different dispersal parameters), creating different types of diversity maps, and conducting the prioritization. The climate, land cover, emissions scenarios, and time horizons could be included here as they are, or included as a separate table.

This figure was created upon suggestions from reviewer 1. He specified the need of such figure in the appendix to complement the summary of the methods from the main text. We followed the recommendation and agreed that such figure is needed for readers more interested in technical details, but not necessarily for the more general audience. They have therefore kept the method network in the appendix.

(23) Table S2 – I note in your response that the acronyms for data sources are described, but I don't see where. It would be helpful to point the reader to that place somewhere in Table S2 (in the caption, or as a note below the table).

The acronyms are “described” by their added sources. The letters were only used as IDs in doing the analyses.

Methods

(24) Climate data – CHELSA CMIP6 climatologies are available for 30 year time slices; for the future periods, should this read 2041-2070 and 2071-2100, in line with what is available at the link provided? Also, only 5 GCMs are available at that website, so it would be helpful to note how MIROC6, AWI-CM-1-1-MR, EC-Earth3, INM-CM5-0 were obtained (or how others wanting to reproduce this analysis can access these data).

We know realized that this could have been better explained. We better described L371-377 how we extracted this CHELSA data not available on the main portal.

(25) Systematic conservation planning, Zonation – Some of the detail in this section should be included in the main text. In particular, details from the last paragraph that discuss the number of scenarios, the included parameters, and the various Zonation algorithms I think are central enough to the paper to warrant inclusion in the main text in the local and regional conservation prioritization sections.

We understand that it sounds tempting to repeat in the main text some details of the methods section; however, we think that this would impend the readability of the manuscript and make unnecessary repetitions more likely to confuse the reader.

(26) Systematic conservation planning, post-analyses – I’m not sure I follow why the top 20% was chosen. Is this because currently only 10% of the study area is included in IUCN I-II and Natura 2000 sites, and an additional 20% would bring the total to 30% in line with Target 3 of the GBF?

We realized that the information could have been better given. So for each expansion scenario (current, 2050 and 2080), the top 20% was chosen because their overlaps extends the network from ~18 to ~35% of the European alps’ total area. We improved the explanation in the legend of figure 4, and L230-234.

Reviewer #3 (Remarks to the Author):

(27) I appreciate a lot the authors’ efforts in revising the manuscript with my previous comments. The manuscript has been improved to a large extent regarding the description of parameters used, the discussions on uncertainties associated with the land cover change, and associated with latitudinal and elevational dispersal. However, I do have some reservations regarding the lack of quantitative

50uncertainty assessment associated with selection of a few key parameter values.

While I agree with the authors that it is not realistic to make all key parameters species specific, the current treatment of these parameter values is not informative to readers. I suggest the authors to quantitatively evaluate the robustness of their results based on sensitivity tests to parameter values within the reasonable range.

Thank you for your positive feedbacks. Regarding the dispersal parameters, we are not sure we understand the question. As previously stated, all parameters were chosen according to existing literature and based on ecological justifications. Employing different parameters would go against these justifications and add more uncertainty/stochasticity to our model outputs. The maturity of the plant is the only unknown factor, but as explained in the discussion (L308-313), decreasing it would only accelerate plant migration, therefore increasing diversity at higher elevation. Employing such migration models (as SDMs) always include uncertainty on how fast changes occur, and this why we are discussing those limitations in the last section.

Our references:

Alexander, J. M., Diez, J. M., & Levine, J. M. (2015). Novel competitors shape species' responses to climate change. *Nature*, 525(7570), 515–518. <https://doi.org/10.1038/nature14952>

Allan, J. R., Possingham, H. P., Atkinson, S. C., Waldron, A., Marco, M. Di, Butchart, S. H. M., Adams, V. M., Kissling, W. D., Worsdell, T., Sandbrook, C., Gibbon, G., Kumar, K., Mehta, P., Maron, M., Williams, B. A., Jones, K. R., Wintle, B. A., Reside, A. E., & Watson, J. E. M. (2022). The minimum land area requiring conservation attention to safeguard biodiversity. *Science*, 376, 1094–1101. <https://doi.org/10.1126/science.abl9127>

Brum, F. T., Graham, C. H., Costa, G. C., Hedges, S. B., Penone, C., Radeloff, V. C., Rondinini, C., Loyola, R., & Davidson, A. D. (2017). Global priorities for conservation across multiple dimensions of mammalian diversity. *Proceedings of the National Academy of Sciences of the United States of America*, 114(29), 7641–7646. <https://doi.org/10.1073/pnas.1706461114>

- Choler, P., Michalet, R., & Callaway, R. M. (2001). Facilitation and competition on gradients in alpine plant communities. *Ecology*, 82(12), 3295–3308. [https://doi.org/10.1890/0012-9658\(2001\)082\[3295:FACOGI\]2.0.CO;2](https://doi.org/10.1890/0012-9658(2001)082[3295:FACOGI]2.0.CO;2)
- Jung, M., Arnell, A., de Lamo, X., García-Rangel, S., Lewis, M., Mark, J., Merow, C., Miles, L., Ondo, I., Pironon, S., Ravilious, C., Rivers, M., Schepaschenko, D., Tallwin, O., van Soesbergen, A., Govaerts, R., Boyle, B. L., Enquist, B. J., Feng, X., ... Visconti, P. (2021). Areas of global importance for conserving terrestrial biodiversity, carbon and water. *Nature Ecology and Evolution*, 5(11), 1499–1509. <https://doi.org/10.1038/s41559-021-01528-7>
- McGowan, J., Beaumont, L. J., Smith, R. J., Chauvenet, A. L. M., Harcourt, R., Atkinson, S. C., Mittermeier, J. C., Esperon-Rodriguez, M., Baumgartner, J. B., Beattie, A., Dudaniec, R. Y., Grenyer, R., Nipperess, D. A., Stow, A., & Possingham, H. P. (2020). Conservation prioritization can resolve the flagship species conundrum. *Nature Communications*, 11(1). <https://doi.org/10.1038/s41467-020-14554-z>
- O'Connor, L. M. J., Pollock, L. J., Renaud, J., Verhagen, W., Verburg, P. H., Lavore, S., Maiorano, L., & Thuiller, W. (2021). Balancing conservation priorities for nature and for people in Europe. *Science*, 373(May), 856–860. <https://doi.org/10.1017/9781108973236.002>
- Pollock, L. J., Thuiller, W., & Jetz, W. (2017). Large conservation gains possible for global biodiversity facets. *Nature*, 546(7656), 141–144. <https://doi.org/10.1038/nature22368>
- Soto-Navarro, C., Ravilious, C., Arnell, A., De Lamo, X., Harfoot, M., Hill, S. L. L., Wearn, O. R., Santoro, M., Bouvet, A., Mermoz, S., Le Toan, T., Xia, J., Liu, S., Yuan, W., Spawn, S. A., Gibbs, H. K., Ferrier, S., Harwood, T., Alkemade, R., ... Kapos, V. (2020). Mapping co-benefits for carbon storage and biodiversity to inform conservation policy and action. *Philosophical Transactions of the Royal Society B: Biological Sciences*, 375(1794). <https://doi.org/10.1098/rstb.2019.0128>
- Vitt, P., Taylor, A., Rakosy, D., Kreft, H., Meyer, A., Weigelt, P., & Knight, T. M. (2023). Global conservation prioritization for the Orchidaceae. *Scientific Reports*, 13(1). <https://doi.org/10.1038/s41598-023-30177-y>

Decision Letter, second revision:

26th October 2023

Dear Dr. Chauvier-Mendes,

52Thank you for submitting your revised manuscript "Transnational conservation to anticipate future plant shifts in Europe" (NATECOLEVOL-23030689B). It has now been seen again by the original Reviewer 2; their comments are below. The reviewer finds that the paper has improved in revision, and therefore we'll be happy in principle to publish it in Nature Ecology & Evolution, pending minor revisions to satisfy the reviewer's final suggestions and to comply with our editorial and formatting guidelines.

[REDACTED]

Reviewer #2 (Remarks to the Author):

Response to Reviewers

Thank you for the opportunity to review the manuscript "Transnational conservation to anticipate future plant shifts in Europe" a second time.

The authors have continued to consider my and the author reviewers' suggestions seriously, and to work to improve the manuscript. The authors' responses to my comments were all addressed adequately, though I would like to reiterate one of my comments that was perhaps unclear and which, if addressed during a further revision, would improve the manuscript and perhaps make it more useful in practice. I would leave it up to the authors to determine whether they agree and would like to add a sentence or two in the conclusions of the manuscript.

My comment from before read: "The authors still have not put the findings and conclusions of this study into the context of the many other regional and global conservation prioritizations out there that focus on different types of biodiversity and different conservation objectives that are all very relevant for 30x30 planning. I think a large proportion of readers, as well as the target audience and general stakeholders in this area would be interested in knowing how the locations of priorities from this study may be aligned with other priorities, and where there might be important tradeoffs to consider."

The reason I'm suggesting more actively acknowledging other global and regional prioritizations that have included the Alps (e.g., Allan et al., 2022; Brum et al., 2017; Jung et al., 2021; McGowan et al., 2020; O'Connor et al., 2021; Pollock et al., 2017; Soto-Navarro et al., 2020; Vitt et al., 2023 as per the authors' response to this comment) is that they very likely do not show similar priority areas (e.g.,

53Switzerland appears to be completely absent per the authors' response). This is likely due to differences in the spatial scale, species, ecosystems, conservation features, etc. considered in the different prioritization studies, but it is important to note that conservation practitioners and policymakers use the results from those studies to guide conservation initiatives and investments.

The current study could make a strong case that such studies do not adequately identify priorities for a robust network of protected areas in the Alps (e.g., by omitting Switzerland), which was identified as having high conservation value in this study. Furthermore, the current study can highlight what is at stake for countries omitting the priority areas in conservation networks that have been reported by other papers in developing 30x30 spatial plans. Doing so would provide readers (including those from governments with decision-making authority) with a more informed and stronger rationale for considering the proposed priority areas for conservation stemming from this paper.

Otherwise, I thank the authors again for a very thorough set of revisions that has led to an improved manuscript.

Our ref: NATECOLEVOL-23030689B

3rd November 2023

Dear Dr. Chauvier-Mendes,

Thank you for your patience as we've prepared the guidelines for final submission of your Nature Ecology & Evolution manuscript, "Transnational conservation to anticipate future plant shifts in Europe" (NATECOLEVOL-23030689B). Please carefully follow the step-by-step instructions provided in the attached file, and add a response in each row of the table to indicate the changes that you have made. Please also check and comment on any additional marked-up edits we have proposed within the text. Ensuring that each point is addressed will help to ensure that your revised manuscript can be swiftly handed over to our production team.

****We would like to start working on your revised paper, with all of the requested files and forms, as soon as possible (preferably within two weeks). Please get in contact with us immediately if you anticipate it taking more than two weeks to submit these revised files.****

When you upload your final materials, please include a point-by-point response to any remaining

54reviewer comments.

In recognition of the time and expertise our reviewers provide to Nature Ecology & Evolution's editorial process, we would like to formally acknowledge their contribution to the external peer review of your manuscript entitled "Transnational conservation to anticipate future plant shifts in Europe". For those reviewers who give their assent, we will be publishing their names alongside the published article.

Nature Ecology & Evolution offers a Transparent Peer Review option for new original research manuscripts submitted after December 1st, 2019. As part of this initiative, we encourage our authors to support increased transparency into the peer review process by agreeing to have the reviewer comments, author rebuttal letters, and editorial decision letters published as a Supplementary item. When you submit your final files please clearly state in your cover letter whether or not you would like to participate in this initiative. Please note that failure to state your preference will result in delays in accepting your manuscript for publication.

Cover suggestions

We welcome submissions of artwork for consideration for our cover. For more information, please see our https://www.nature.com/documents/Nature_covers_author_guide.pdf guide for cover artwork.

Nature Ecology & Evolution has now transitioned to a unified Rights Collection system which will allow our Author Services team to quickly and easily collect the rights and permissions required to publish your work. Approximately 10 days after your paper is formally accepted, you will receive an email in providing you with a link to complete the grant of rights. If your paper is eligible for Open Access, our Author Services team will also be in touch regarding any additional information that may be required to arrange payment for your article.

Please note that *Nature Ecology & Evolution* is a Transformative Journal (TJ). Authors may publish their research with us through the traditional subscription access route or make their paper immediately open access through payment of an article-processing charge (APC). Authors will not be required to make a final decision about access to their article until it has been accepted. <https://www.springernature.com/gp/open-research/transformative-journals> Find out more

55about Transformative Journals

Authors may need to take specific actions to achieve [compliance with funder and institutional open access mandates](https://www.springernature.com/gp/open-research/funding/policy-compliance-faqs). If your research is supported by a funder that requires immediate open access (e.g. according to [Plan S principles](https://www.springernature.com/gp/open-research/plan-s-compliance)) then you should select the gold OA route, and we will direct you to the compliant route where possible. For authors selecting the subscription publication route, the journal's standard licensing terms will need to be accepted, including [self-archiving and license to publish](https://www.nature.com/nature-portfolio/editorial-policies/self-archiving-and-license-to-publish). Those licensing terms will supersede any other terms that the author or any third party may assert apply to any version of the manuscript.

[REDACTED]

[REDACTED]

Reviewer #2:
Remarks to the Author:
Response to Reviewers

Thank you for the opportunity to review the manuscript "Transnational conservation to anticipate future plant shifts in Europe" a second time.

The authors have continued to consider my and the author reviewers' suggestions seriously, and to work to improve the manuscript. The authors' responses to my comments were all addressed adequately, though I would like to reiterate one of my comments that was perhaps unclear and which, if addressed during a further revision, would improve the manuscript and perhaps make it more useful in practice. I would leave it up to the authors to determine whether they agree and would like to add a sentence or two in the conclusions of the manuscript.

My comment from before read: "The authors still have not put the findings and conclusions of this study into the context of the many other regional and global conservation prioritizations out there that

56focus on different types of biodiversity and different conservation objectives that are all very relevant for 30x30 planning. I think a large proportion of readers, as well as the target audience and general stakeholders in this area would be interested in knowing how the locations of priorities from this study may be aligned with other priorities, and where there might be important tradeoffs to consider.”

The reason I’m suggesting more actively acknowledging other global and regional prioritizations that have included the Alps (e.g., Allan et al., 2022; Brum et al., 2017; Jung et al., 2021; McGowan et al., 2020; O’Connor et al., 2021; Pollock et al., 2017; Soto-Navarro et al., 2020; Vitt et al., 2023 as per the authors’ response to this comment) is that they very likely do not show similar priority areas (e.g., Switzerland appears to be completely absent per the authors’ response). This is likely due to differences in the spatial scale, species, ecosystems, conservation features, etc. considered in the different prioritization studies, but it is important to note that conservation practitioners and policymakers use the results from those studies to guide conservation initiatives and investments.

The current study could make a strong case that such studies do not adequately identify priorities for a robust network of protected areas in the Alps (e.g., by omitting Switzerland), which was identified as having high conservation value in this study. Furthermore, the current study can highlight what is at stake for countries omitting the priority areas in conservation networks that have been reported by other papers in developing 30x30 spatial plans. Doing so would provide readers (including those from governments with decision-making authority) with a more informed and stronger rationale for considering the proposed priority areas for conservation stemming from this paper.

Otherwise, I thank the authors again for a very thorough set of revisions that has led to an improved manuscript.

Final Decision Letter:

22nd November 2023

Dear Dr Chauvier-Mendes,

We are pleased to inform you that your Article entitled "Transnational conservation to anticipate future plant shifts in Europe", has now been accepted for publication in Nature Ecology & Evolution.

Over the next few weeks, your paper will be copyedited to ensure that it conforms to Nature Ecology and Evolution style. Once your paper is typeset, you will receive an email with a link to choose the appropriate publishing options for your paper and our Author Services team will be in touch regarding any additional information that may be required

Due to the importance of these deadlines, we ask you please us know now whether you will be difficult

57to contact over the next month. If this is the case, we ask you provide us with the contact information (email, phone and fax) of someone who will be able to check the proofs on your behalf, and who will be available to address any last-minute problems. Once your paper has been scheduled for online publication, the Nature press office will be in touch to confirm the details.

Acceptance of your manuscript is conditional on all authors' agreement with our publication policies (see www.nature.com/authors/policies/index.html). In particular your manuscript must not be published elsewhere and there must be no announcement of the work to any media outlet until the publication date (the day on which it is uploaded onto our web site).

Please note that *Nature Ecology & Evolution* is a Transformative Journal (TJ). Authors may publish their research with us through the traditional subscription access route or make their paper immediately open access through payment of an article-processing charge (APC). Authors will not be required to make a final decision about access to their article until it has been accepted. [Find out more about Transformative Journals](https://www.springernature.com/gp/open-research/transformative-journals)

Authors may need to take specific actions to achieve [compliance with funder and institutional open access mandates](https://www.springernature.com/gp/open-research/funding/policy-compliance-faqs). If your research is supported by a funder that requires immediate open access (e.g. according to [Plan S principles](https://www.springernature.com/gp/open-research/plan-s-compliance)) then you should select the gold OA route, and we will direct you to the compliant route where possible. For authors selecting the subscription publication route, the journal's standard licensing terms will need to be accepted, including [those licensing terms](https://www.nature.com/nature-portfolio/editorial-policies/self-archiving-and-license-to-publish) will supersede any other terms that the author or any third party may assert apply to any version of the manuscript.

We welcome the submission of potential cover material (including a short caption of around 40 words) related to your manuscript; suggestions should be sent to Nature Ecology & Evolution as electronic files (the image should be 300 dpi at 210 x 297 mm in either TIFF or JPEG format). Please note that such pictures should be selected more for their aesthetic appeal than for their scientific content, and

58that colour images work better than black and white or grayscale images. Please do not try to design a cover with the Nature Ecology & Evolution logo etc., and please do not submit composites of images related to your work. I am sure you will understand that we cannot make any promise as to whether any of your suggestions might be selected for the cover of the journal.

You can generate the link yourself when you receive your article DOI by entering it here: <http://authors.springernature.com/share>.

Thank you again for choosing NEE to publish your work; I look forward to seeing it published soon.

[REDACTED]

P.S. Click on the following link if you would like to recommend Nature Ecology & Evolution to your librarian <http://www.nature.com/subscriptions/recommend.html#forms>

** Visit the Springer Nature Editorial and Publishing website at http://editorial-jobs.springernature.com?utm_source=ejp_NEcoE_email&utm_medium=ejp_NEcoE_email&utm_campaign=ejp_NEcoE for more information about our career opportunities. If you have any questions please click [here](mailto:editorial.publishing.jobs@springernature.com). **